# The *Helicobacter pylori* Genome Project: insights into *H. pylori* population structure from analysis of a worldwide collection of complete genomes

Kaisa Thorell [1,204] ✉, Zilia Y. Muñoz-Ramírez [2,204], Difei Wang[3,4], Santiago Sandoval-Motta[5,6,7], Rajiv Boscolo Agostini[8], Silvia Ghirotto[8], Roberto C. Torres [9], HpGP Research Network*, Daniel Falush [9], M. Constanza Camargo [4,205] & Charles S. Rabkin[4,205]

*Helicobacter pylori*, a dominant member of the gastric microbiota, shares co-evolutionary history with humans. This has led to the development of genetically distinct *H. pylori* subpopulations associated with the geographic origin of the host and with differential gastric disease risk. Here, we provide insights into *H. pylori* population structure as a part of the *Helicobacter pylori* Genome Project (*Hp*GP), a multi-disciplinary initiative aimed at elucidating *H. pylori* pathogenesis and identifying new therapeutic targets. We collected 1011 well-characterized clinical strains from 50 countries and generated high-quality genome sequences. We analysed core genome diversity and population structure of the *Hp*GP dataset and 255 worldwide reference genomes to outline the ancestral contribution to Eurasian, African, and American populations. We found evidence of substantial contribution of population hpNorthAsia and subpopulation hspUral in Northern European *H. pylori*. The genomes of *H. pylori* isolated from northern and southern Indigenous Americans differed in that bacteria isolated in northern Indigenous communities were more similar to North Asian *H. pylori* while the southern had higher relatedness to hpEastAsia. Notably, we also found a highly clonal yet geographically dispersed North American subpopulation, which is negative for the *cag* pathogenicity island, and present in 7% of sequenced US genomes. We expect the *Hp*GP dataset and the corresponding strains to become a major asset for *H. pylori* genomics.

*Helicobacter pylori* has co-existed with humans for more than 100,000 years. It is the primary etiologic agent associated with gastric diseases such as ulcers and gastric cancer. Still, while over half the world population is colonized with *H. pylori*, less than 2% will end up with gastric cancer[1,2]. The intimate symbiotic relationship with humans,

together with predominantly vertical transmission, has led *H. pylori* to evolve into multiple distinct geographic populations[3–5]. The phylo-geographic structure of *H. pylori* is classified into major populations ("hp") and subpopulations ("hsp") that correlate with ancient human migrations[3,4,6]. However, most worldwide efforts in this regard have

A full list of affiliations appears at the end of the paper. *A list of authors and their affiliations appears at the end of the paper. ✉e-mail: kaisa.thorell@gu.se

been based on the analysis of only a handful of genes rather than whole genomes[3,4]. The risk of developing disease from *H. pylori* infection varies greatly by geography[7] and genomic studies of both humans and *H. pylori* are required to identify the factors that modify this risk.

The *Helicobacter pylori* Genome Project (*Hp*GP) is an international and multidisciplinary initiative to sequence and map *H. pylori* population structure by collecting strains worldwide. Here we analyze 1011 *H. pylori* genomes, sequenced with PacBio Single Molecule, Real-Time long-read technology, which made it possible to acquire complete assemblies. By relating the *Hp*GP dataset to a reference set of known population assignment, we were able to quantify, with great resolution, the different inferred ancestral sources of *H. pylori* subpopulations and the recent and ongoing admixture among subpopulations.

## Results

### *Hp*GP is a dataset of high quality and worldwide representation

The *Hp*GP has assembled clinical strains from 50 countries, including 12 countries from which no *H. pylori* genome sequences have previously been published (Table 1). Out of the 1011 genomes, all but seven were completely circularized (Supplementary Data 1).

To investigate the population structure of the *Hp*GP dataset, we performed fineSTRUCTURE (FS), chromosome painting, and network analyses of shared core genome features as described[8], and discriminant analysis of principal components (DAPC)[9]. To anchor the dataset, we used 255 *H. pylori* reference genomes with known Hp/hsp population assignments, representing 17 global subpopulations (Supplementary Data 2). In total, the core genome (set of homologous genes present in >95% of genomes) of the *Hp*GP dataset, the *Hp*GP-26695 reference genome, and 255 worldwide references, consisted of 1227 genes.

The fineSTRUCTURE global analysis revealed four main *H. pylori* population clusters: (i) Southwest Europe, including Latin America and Northeast Africa, (ii) Northern and Central Europe, Middle East, and Central Asia, (iii) Western and Southern Africa, including Africa2 and North, South and Central America, and (iv) North, Central and East Asia, and Indigenous populations in America. In total, these formed 17 main subpopulations (Fig. 1 and Supplementary Figs. 1 and 2). The network and DAPC analyses supported this structure but with six main clusters of differentiation (Fig. 2 and Supplementary Fig. 3).

South Africa (DAPC group 4, hpAfrica2 in FS) and the reference genomes from Australia/New Guinea (DAPC group 6, hpSahul in FS) differentiated extensively from the others. A further DAPC analysis not considering these two groups showed a clear separation of two of the remaining clusters from the others: one composed of isolates of African and American origin (DAPC group 2, FS cluster III) and one that includes isolates from Central/East Asia and Indigenous Americans (DAPC group 5, FS cluster IV). The remaining (groups 1 and 3) were more similar and intertwined, representing Southern Europe/Northeast Africa and Eurasia/Central Asia and Americas, respectively (Supplementary Fig. 3). The population assignments according to the respective analyses are summarized in Supplementary Data 3.

### The hpEurope subpopulations span from the Atlantic coast to South Asia

In the fineSTRUCTURE analysis, three main European/Eurasian subpopulations emerged (Supplementary Figs. 1 and 2), of which hspNEurope and hspSWEurope have previously been described[8,10]. The hspEurasia population is proposed in this study, and includes the already reported hspCEurope/hspSEurope[8,10–12], and hspMiddleEast[10]. Previous studies had limited coverage of Eastern Europe and the Middle East. The *Hp*GP strains from Lithuania, Latvia, Russia, Poland, Bulgaria, Türkiye, and Jordan allowed mapping of the Eurasian *H. pylori* relationships with unprecedented detail (Fig. 1). Two northern European populations showed an east-west differentiation, in

**Table 1 | Summary of the *Hp*GP strain collection**

| Country | Total number | Non-atrophic gastritis (%) | Intestinal metaplasia (%) | Gastric cancer (%) |
|---|---|---|---|---|
| Algeria[a] | 10 | 100 | | |
| Argentina | 10 | 100 | | |
| Bangladesh | 10 | 100 | | |
| Brazil | 21 | 48 | 38 | 14 |
| Bulgaria[a] | 8 | 100 | | |
| Canada | 20 | 35 | 65 | |
| Chile | 46 | 54 | 46 | |
| China | 10 | | | 100 |
| DR Congo[a] | 11 | 91 | | 9 |
| Colombia | 45 | 78 | 16 | 7 |
| Costa Rica | 8 | 100 | | |
| Dominican Republic[a] | 11 | 91 | | 9 |
| France | 21 | 48 | | 52 |
| Germany | 17 | 59 | | 41 |
| Ghana[a] | 2 | 100 | | |
| The Gambia | 5 | 100 | | |
| Greece | 21 | 48 | | 52 |
| Guatemala | 3 | 100 | | |
| Honduras | 26 | 35 | 38 | 27 |
| India | 10 | 100 | | |
| Indonesia | 11 | 91 | | 9 |
| Iran | 4 | 100 | | |
| Iceland[a] | 11 | 91 | 9 | |
| Israel | 10 | 70 | 30 | |
| Italy | 29 | 34 | 34 | 31 |
| Japan | 29 | 38 | 21 | 41 |
| Jordan[a] | 10 | 100 | | |
| Kazakhstan[a] | 2 | 100 | | |
| Kyrgyzstan[a] | 10 | 100 | | |
| Korea | 54 | 19 | 19 | 63 |
| Latvia[a] | 34 | 29 | 24 | 47 |
| Lithuania | 23 | 43 | 35 | 22 |
| Malaysia | 19 | 47 | 53 | |
| Mexico | 22 | 45 | | 55 |
| Myanmar[a] | 12 | 83 | | 17 |
| Nepal | 13 | 77 | | 23 |
| Nigeria | 4 | 100 | | |
| Peru | 33 | 30 | 24 | 45 |
| Poland[a] | 20 | 100 | | |
| Portugal | 30 | 57 | 27 | 17 |
| Russia | 10 | 60 | | 40 |
| Singapore | 21 | 38 | 33 | 29 |
| South Africa | 9 | 100 | | |
| Spain | 106 | 72 | 13 | 15 |
| Sweden | 30 | 33 | 33 | 33 |
| Switzerland | 15 | 60 | 40 | |
| Taiwan | 24 | 42 | | 58 |
| Türkiye | 17 | 59 | 18 | 24 |
| US (continental) | 68 | 96 | 1 | 3 |
| US (Puerto Rico)[a] | 7 | 100 | | |
| Vietnam | 9 | | | 100 |
| Total | 1011 | 60 | 17 | 23 |

[a]Geographical areas from which no *H. pylori* whole-genome sequences were previously available in GenBank

hspNEurope an east clade with genomes from Latvia, Lithuania, and Russia separated from a north-western clade with genomes from UK, Sweden, Iceland, and Canada (Supplementary Fig. 2). Within hspEurasia, three main clades could be noted of which two spanned from west

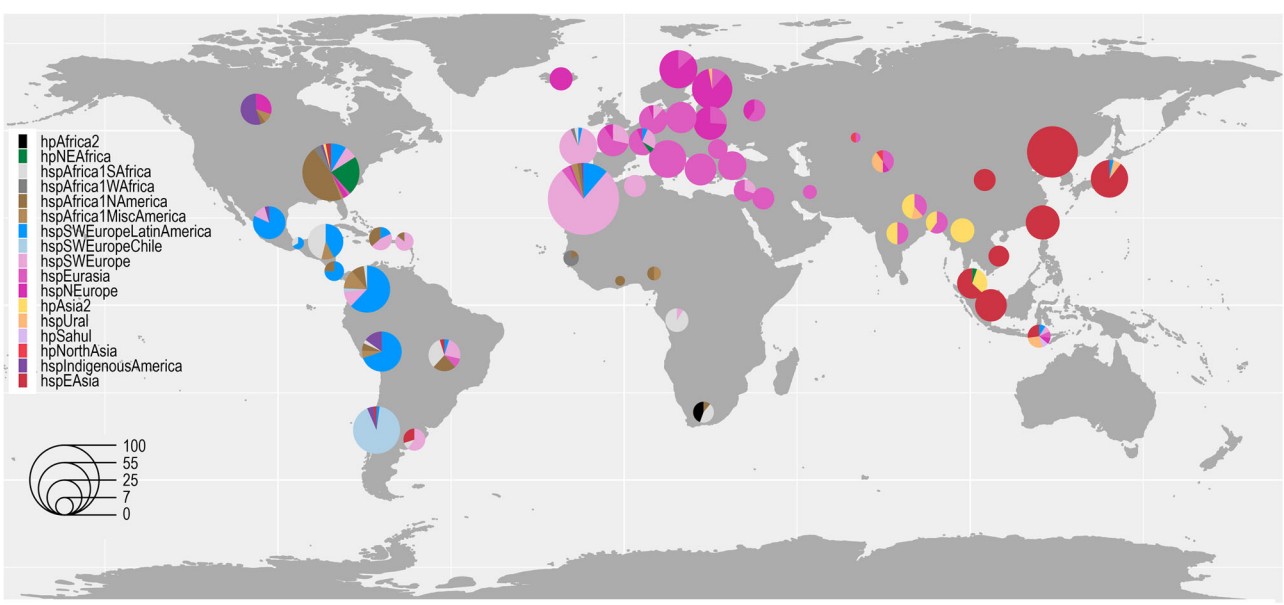

**Fig. 1 | World map of *Hp*GP strain origins and population assignments.** The area of each pie is proportional to the number of *Hp*GP genomes from each country and colored by the *H. pylori* population (hp) and subpopulation (hsp) as assigned by fineSTRUCTURE (Supplementary Figs. 1 and 2).

to east (Supplementary Fig. 2). The first, Central-Eastern European hspEurasia1, dominates in Germany, Poland, Lithuania, Latvia, Türkiye, and Russia, while hspEurasia2 is more Southern with representation from France in the west, via Italy and Greece, to Jordan and Iran in the Middle East. Thirdly, hspEurasia3 includes genomes from India and Bangladesh, but also Greece, which separated from the others but were still within the hspEurasia subpopulation.

## The European subpopulations have different ancestry proportions

To further investigate the proposed subpopulations we inferred ancestry by comparing genomes within our contemporary dataset in a directed chromosome painting using only the proposed *H. pylori* ancestral populations hpAfrica2, hpNEafrica, hspAfrica1WAfrica, hpAsia2, hspUral, hpNorthAsia, and hspEAsia as donors (i.e., contributors of genomic ancestry)[3,10]. We confirmed a gradient in inferred ancestry along both the north-south axis with increasing Asian ancestry and decreasing African ancestry in the hspEurasia1 and hspNEurope populations and the east-west axis with hspSWEurope having a higher proportion of hspAfrica1WAfrica ancestry and with the similar contribution of hpNEafrica as the Eurasia2 population (Fig. 3 and Supplementary Fig. 3).

The more central Asian hspEurasia3 on the other hand, showed markedly higher hpAsia2 ancestry than the other hpEurope populations, concordant with its geographical co-existence with hpAsia2. Interestingly, hspUral was a more pronounced Asian ancestor for all the hpEurope subpopulations than hpNorthAsia and hspEAsia, the latter two being very even contributors, except for in hspNEurope, where hpNorthAsian ancestry was slightly higher. This relationship was also supported by the network analysis (Fig. 2).

## Central Asia can be described with increased resolution but still has underrepresented regions

Apart from the relatively well-investigated hspEAsian subpopulation[13], the fineSTRUCTURE analysis grouped the central Eurasian strains into three main clades: hpAsia2 and two clades preliminarily termed hpNorthAsia and hspUral, based on their association with reference strains previously described by Moodley et al.[14].

HpAsia2 is one of the main ancestral populations of *H. pylori* but has been comparatively understudied. In the *Hp*GP dataset, genomes belonging to hpAsia2 are mainly from India, Bangladesh, Myanmar, and Nepal, with the Nepalese forming a clade slightly separated from the others (Supplementary Fig. 2a, c). As seen in Fig. 1, hpAsia2 co-exists with the hpEurope hspEurasia3 population in all these countries, except for Myanmar, where only hpAsia2 is present. The DAPC analysis, on the other hand, did not distinguish hpAsia2 from hspNEurope and hspEurasia using $k = 6$, while the separation was evident and very consistent using $k = 17$ (Supplementary Fig. 4).

HpNorthAsia was previously established as one of the main Siberian populations using Multilocus Sequence Typing (MLST)[14]. In our reference panel, hpNorthAsia (including hspAltai) and its subpopulation hspSiberia1 were represented by genomes from central and eastern Siberia. In our analyses, these two populations did not segregate, and *Hp*GP genomes from Kazakhstan and Kyrgyzstan were also associated with this cluster (Fig. 1 and Supplementary Fig. 1).

hspUral has been suggested as a southern central Asian subpopulation of hpAsia2. In our dataset, a cluster with a relatively wide geographical representation from Kazakhstan and Kyrgyzstan to Indonesia and Japan (Fig. 1 and Supplementary Figs. 1 and 2) is associated with the hspUral reference genomes. Our main chromosome painting analysis suggested the proposed hspUral population to contain two subclades with very different painting profiles (Supplementary Fig. 2), which was supported by the DAPC and network analysis, and the fineSTRUCTURE principal component analysis (PCA) (see https://hpgp.shinyapps.io/Interactive_figures, Fig. 4). The ancestral contributions to the central Asian genomes confirmed the *Hp*GP "hspUral" clade not to have pronounced contribution by the hspUral references but relatively high hpAsia2, hpNorthAsia and hspEAsia painting proportions (Fig. 3). The variability of contributions was also high within the clade, suggesting this may not constitute one pure subpopulation but may consist of representatives of several HpAsia subpopulations (https://hpgp.shinyapps.io/Interactive_figures, Fig. 2). One hpAsia2 reference genome, L7, from Ladakh in northern India grouped with this cluster, especially close to two Nepalese genomes. Several "hspUral" genomes also showed an association with hpSahul in the chromosome painting (Supplementary Fig. 2), which may indicate a relationship between this group and the recently suggested hpRyukyu[15].

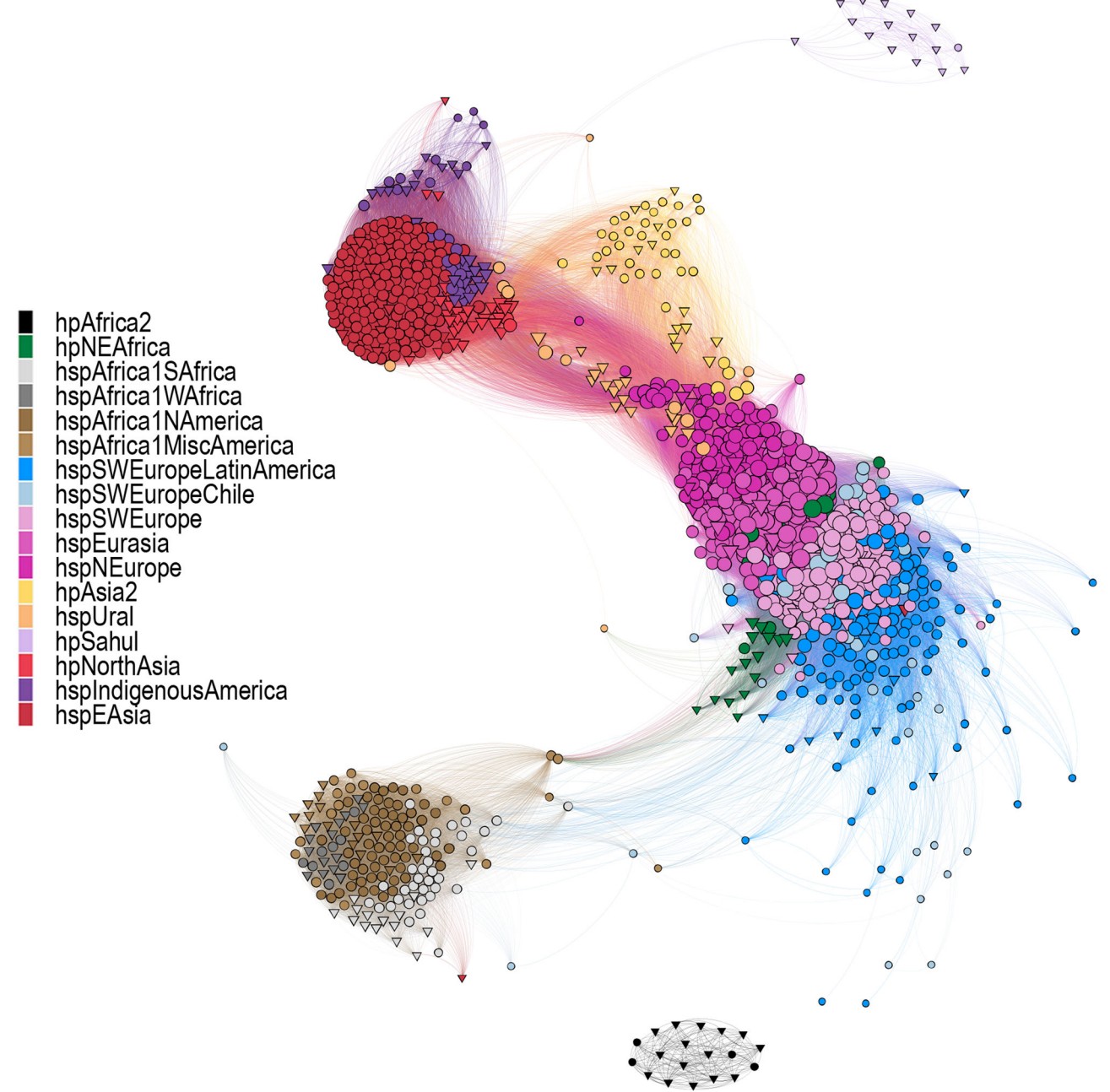

**Fig. 2 | Distance network analyses of the core genome of the *H. pylori* strains studied.** Fruchterman–Reingold layout of the pruned distance network between *Hp*GP genomes (circles) and reference genomes (triangles) (see Methods). Colors indicate the *H. pylori* population (hp) and subpopulation (hsp) as assigned by fineSTRUCTURE (Supplementary Figs. 1 and 2). The length and opacity of each link are proportional to the genetic distance between genomes (nodes), with higher opacity and shorter length indicating genetic closeness and less opacity and higher length indicating higher genetic distance between strains. The size of each node is proportional to the connectivity (number of links) of that node, indicating that bigger nodes have connections to more other strains than those of lesser sizes.

## African and African-descent genomes

The *Hp*GP dataset includes African genomes from understudied countries such as Algeria, Democratic Republic of Congo (DRC), Ghana, and Nigeria, adding to previous knowledge from the Gambia and South Africa. The fineSTRUCTURE analysis confirms earlier observations of the presence of four African populations in this continent, hpAfrica2, in the *Hp*GP dataset represented in South Africa; hspAfrica1SAfrica, which reaches as far north as DRC; hspAfrica1WAfrica represented in the Gambia, as previously reported, and hpNEAfrica. However, the Ghanaian and Nigerian genomes grouped with the more admixed hspAfrica1NorthAmerica and hspAfrica1MiscAmericas populations, interspersed with, and by chromosome painting indistinguishable from

genomes from the US, Puerto Rico (US territory), Dominican Republic, Colombia, and Brazil, likely a result of the trans-Atlantic slave trade from West Africa into the Americas.

The East African reference genomes from Sudan and Ethiopia grouped within the hspSWEurope umbrella but distinctive from the European SWEurope clade, instead forming a cluster with North American genomes (Fig. 1 and Supplementary Fig. 1). In the fineSTRUCTURE PCA plots, especially pronounced in PC10 and PC11, the reference hpNEAfrican genomes and the US *Hp*GP genomes clearly formed two segregated groups except for one Malaysian and one Swiss genome that grouped with the references (https://hpgp.shinyapps.io/ Interactive_figures, Fig. 5). Both the DAPC and network analysis

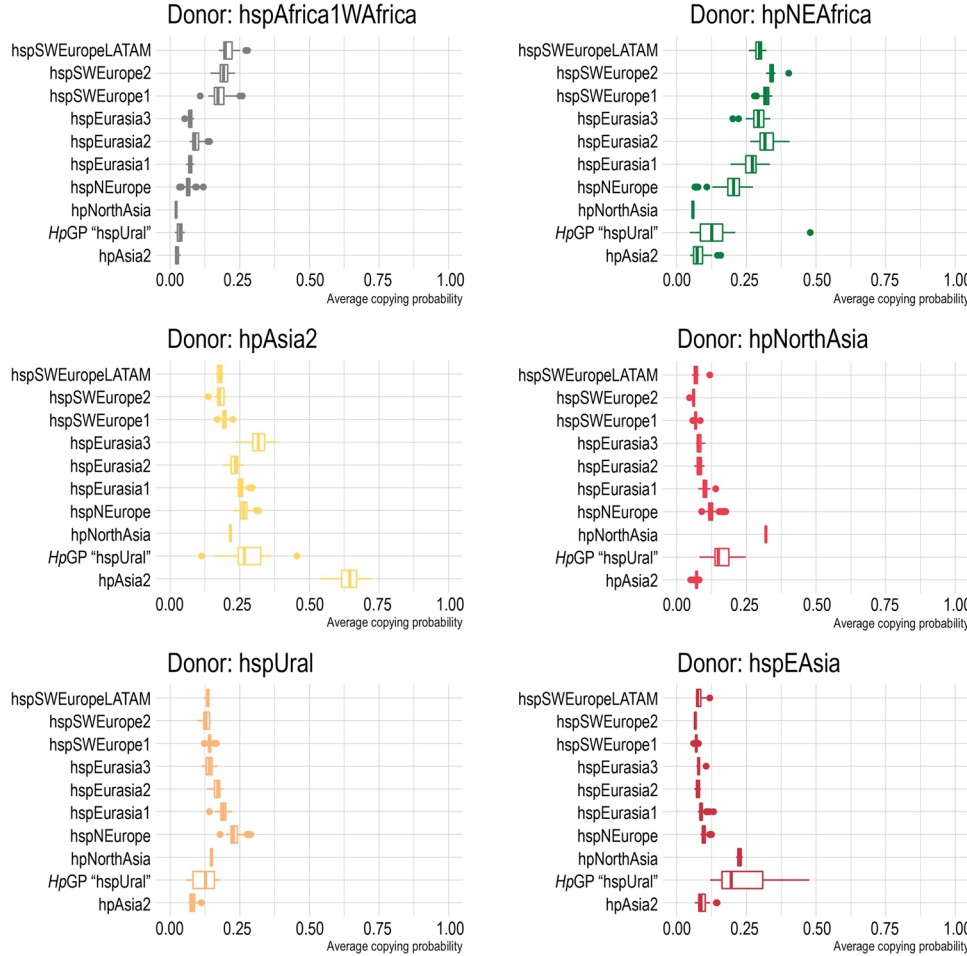

**Fig. 3 | Inferred ancestral genomic contributions to the Eurasian *Hp*GP genomes.** Ancestral chromosome painting proportions by donor and Eurasian subpopulation. Boxplots show the median value per group, and the 25th and 75th percentiles (hinges), with whiskers extending from the hinge to the largest value no further than 1.5 × IQR (inter-quartile range) from the hinge. Data points beyond the whiskers are plotted individually. The number of genomes in each respective Eurasian population is hspSWEuropeLatinAmerica, *n* = 15; hspSWEurope2, *n* = 12; hspSWEurope1, *n* = 129; hspEurasia3, *n* = 18; hspEurasia2, *n* = 76; hspEurasia1, *n* = 103; hspNEurope, *n* = 95; hpNorthAsia, *n* = 2; *Hp*GP "hspUral", *n* = 10; hpAsia2, *n* = 27.

supported the separation of the hpAfrica1 population from hpNEAfrica, the latter being intermingled with genomes from southern Europe and Iberia.

The Algerian strains did not cluster with the other African strains, but within hspSWEurope, together with genomes from Israel and Colombia in a cluster we termed SWEurope2. Despite showing slightly higher West and Northeast African and lower Asian ancestry than SWEurope1 (https://hpgp.shinyapps.io/Interactive_figures, Fig. 2), our analysis confirmed that North African *H. pylori* more closely resemble Iberian and Middle Eastern bacteria than African bacteria.

**North America hosts a geographically dispersed deep clone**
The *Hp*GP dataset contains 68 genomes from the wide geographical representation of the continental US. This feature allowed us to identify a novel subpopulation of 15 US isolates, which showed high similarity and clustered together with genomes of hpNEAfrican ancestry in the fineSTRUCTURE analysis (Supplementary Figs. 1 and 2) and of which none carried the cag pathogenicity island (*cagPAI*).

High levels of sequence homogeneity within *H. pylori* are unexpected as unrelated strains differ in their DNA sequence at almost all genes. To further investigate the novel US subpopulation, we performed core genome (cg) MLST of the entire dataset (Fig. 4a). Within the *Hp*GP, over 64% of strain pairs differ in sequence at all the 1040 genes. Even amongst strains sampled from the same country, 34%

differ in all the genes. Only 0.15%, 798 pairs, shared similarity at >1% of genes. All but 213 of these pairs are between strains in the same country. Nearly a tenth (66) of these pairs is found between a group of 12 US strains, showing allele distances between 0.83 and 0.94 (17–6% identical alleles, respectively). Thus, this group represents older clonal relationships, a putative "deep clone"; a set of strains that share a recent common ancestor but have diverged via homologous recombination at a large fraction of their genome. Three strains are somewhat less related to these 12, sharing between 1% and 7% of genes, and were conservatively excluded from this clonal group. Other pairs involving more than two samples from the same population also showed deep clonal relationships (e.g., hspSWEuropeChile). However, the amount and pattern of alleles shared between these samples could be better explained by genetic drift and further analysis within this population is needed to define the boundaries of a putative clone.

The *Hp*GP strains from the deep clonal group were sampled from California, Wisconsin, Tennessee, Arkansas, Georgia, and Texas and, in total, represented a fifth of the *Hp*GP US genomes. Kmer-based clustering analysis showed an additional five public genomes from two other geographical sources, Ohio and Louisiana, associating closely with the proposed deep clonal group. We used ClonalFrameML to estimate the relationships between the genomes. Assuming a previously estimated $1.38 \times 10^{-5}$ mutation rate per site per year[16], the common ancestor lived an estimated 175 years before the strains were

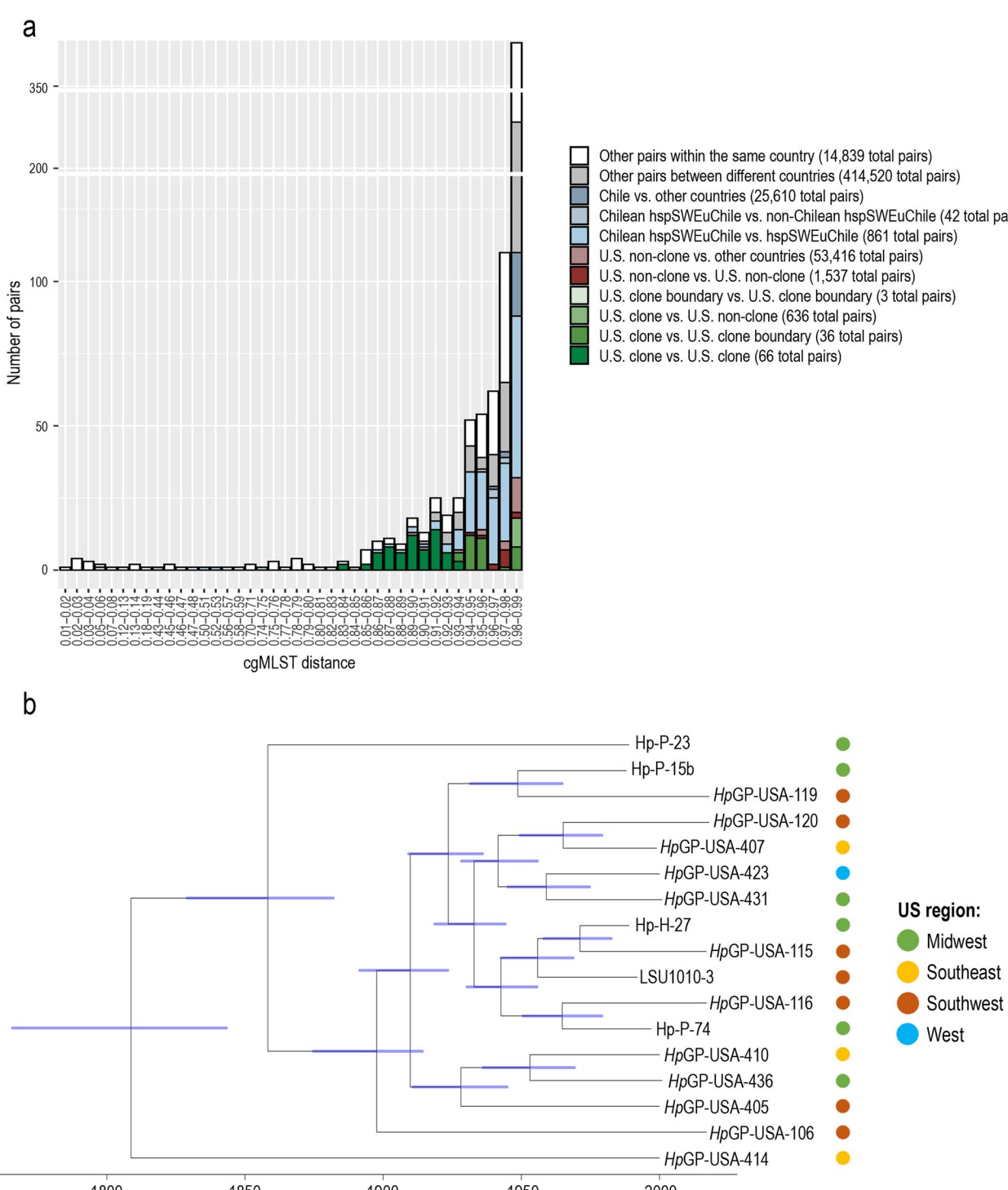

**Fig. 4 | In-depth analysis of clonal relationships in the global *H. pylori* dataset.** **a** Pairwise core genome MLST (cgMLST) distances of the *Hp*GP dataset. Bins illustrate the distribution of core genome allele sharing between pairs of samples. The *x*-axis ranges from 0.1 to 0.99, with lower values indicating higher number of shared alleles. Every pair is included in a single category of comparison (color bar). Only a small fraction of all possible pairs shares more than 1% of alleles, most of them involving samples from the same country of origin. It is noteworthy that a group of strains from different regions of the US shares between 6% and 17% of alleles corresponding to 62 and 176 identical genes, suggesting the presence of a deep clone. Other pairs exhibit larger portions of shared alleles (distances <50%), representing recent transmissions between closely related strains. **b** Dated ClonalFrameML tree of the final set of strains considered to belong to the US deep clone Hp_Clone_US-1, including five publicly available genomes. Node ages correspond to years based on a previously estimated $1.38 \times 10^{-5}$ mutation rate per site per year. The colored dots represent the geographical origin of each strain.

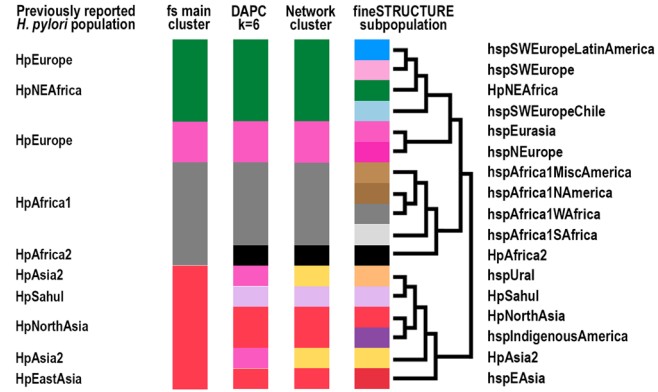

**Fig. 5 | Summary of population classifications.** Summary of the clustering results using the respective analyses in relation to previously reported MLST and whole genome-based *H. pylori* populations (Hp) and subpopulations (hsp). Colors are based on classifications from the fineSTRUCTURE (fs) analyses visualized in Supplementary Fig. 1, on the *K* = 6 discriminant analysis of principal components, DAPC (Supplementary Fig. 3), and the network clusters (Fig. 2). The topology of the dendrogram to the left is based on the fineSTRUCTURE hierarchical clustering of Supplementary Fig. 1.

collected (95% confidence interval, 107–227 years), while the majority of internal nodes are estimated to be less than 50 years old (Fig. 4b). Thus, the sampled strains are not epidemiologically associated with each other, and instead represent independent strains from a circulating population of clonally related bacteria, which we suggest calling Hp_Clone_US-1.

**Latin American subpopulations are more admixed than others**
A total of 238 strains from different regions of Latin America were included in the *Hp*GP (Table 1). In the fineSTRUCTURE analysis, most Latin American strains clustered into two previously described populations, hspAfrica1MiscAmerica and hspSWEuropeLatinAmerica[8,11], and in hspSWEuropeChile (Supplementary Figs. 1 and 2). Around one-third of the Latin American genomes clustered in non-Latin American populations, the majority in hspAfrica1SAfrica, and hspSWEurope. However, there were also hspEAsia genomes in Argentina, Brazil, and Chile and two hspEurasia genomes from Brazil. Generally, the Latin American genomes were more admixed than their European and African counterparts, with a higher African proportion in hspSWEurope Latin American genomes and a higher European proportion in genomes grouping with hspAfrica1 (https://hpgp.shinyapps.io/Interactive_figures, Fig. 3)

Notably, most Chilean isolates clustered in a separate group, hspSWEuropeChile, (Supplementary Fig. 1), similar to Colombian isolates (hspSWEuropeColombia) previously described[8,11]. This population is close to hspSWEuropeLatinAmerica and hspSWEurope, as can be seen in the fineSTRUCTURE PCA, particularly in components PC1 and PC7 (https://hpgp.shinyapps.io/Interactive_figures, Fig. 5). However, in the DAPC and network analyses, these strains are dispersed but still near hspSWEurope (Fig. 2 and Supplementary Fig. 3), which is supported by very high self-painting proportions in the chromosome painting analyses (Supplementary Fig. 2), and high pairwise similarities between the genomes of this subpopulation in the cgMLST analysis (Fig. 4a).

**Indigenous American *H. pylori* have different ancestral contributions**
The fineSTRUCTURE analysis confirmed the hspIndigenousAmerica group[8,14]. This population is made up of isolates from urban areas of mixed human ancestry, as well as Indigenous communities. HspIndigenousAmerica can be subdivided into two groups called

hspIndigenousNAmerica and hspIndigenousSAmerica (Supplementary Figs. 1 and 2). While hspIndigenousNAmerica is composed of strains from Indigenous communities in North America (Canada and US), the hspIndigenousSAmerica group mostly contains isolates from Latin American regions. In this dataset, we added observations of this subpopulation in Chile, Mexico, Peru, Spain, and the US.

According to the ancestral chromosome painting, and corroborated by the network results, hspIndigenousSAmerica shows a higher proximity to hspEAsia, while hspIndigenousNAmerica has a higher Indigenous-ancestral proportion and is closer to hpNorthAsia in the network analysis, even relatively distanced from hspIndigenousSAmerica (Fig. 2, https://hpgp.shinyapps.io/Interactive_figures, Fig. 3).

## Discussion
The intimate association between humans and *H. pylori* started at the beginning of our species and represents a unique story of co-evolution between kingdoms that has fascinated researchers and the public and contributed to understanding human migration dynamics[14,17]. However, the challenge is to understand the consequences of this thousands-of-years of co-evolution for human health, and on the whole-genome level, bacterial population structure has mostly been studied in the setting of specific geographical areas[8,10,13–15,18–20]. Ongoing analyses by the *Hp*GP Research Network are comparing between strains from patients with different gastric diseases in order to identify genetic and epigenetic bacterial features that determine human pathogenicity. The *Hp*GP provides a publicly available worldwide collection of complete genomes and epigenomes with high-quality metadata for future investigations of *H. pylori* pathobiology.

Here we present a phylogeographic characterization of the *Hp*GP genomes and outline the global population structure of this bacterium. We used three complementing comparative genomics approaches, fineSTRUCTURE/Chromosome Painting analysis, DAPC, and network analysis of pairwise distances, including interactive visualization of the data, which allowed us to study different aspects of the genomic relationships. A summary of the classifications using the different methods, including their relation to previously reported populations, is presented in Fig. 5, with details in Supplementary Data 3. The higher dynamic range of the DAPC and network analysis clearly showed that hpAfrica2 and hpSahul were very distant from all other populations (Fig. 2, Supplementary Fig. 3b and https://hpgp.shinyapps.io/Interactive_figures, Fig. 1), and the DAPC presented another four main clusters of similarity: a South/West African cluster and a NEAfrica/SWEurope cluster, of which both also had a high presence in the Americas, a North-Central Eurasian cluster, and a North/East Asian cluster, which also included hspIndigenousAmerica. All analyses, however, additionally provided evidence for strong interactions between the hspEurasia and hspSWEurope genomes, and in the 3D plots of ancestry contribution, these populations form a continuum of different ancestry levels, rather than being discrete populations (https://hpgp.shinyapps.io/Interactive_figures, Figs. 2 and 4). Iterating the DAPC analysis to test the consistency of classifications showed, for example, that northeast European genomes from Latvia, Lithuania, Poland, and Russia interchangeably were classified to the clusters corresponding to hspNEurope and hspEurasia. Similarly, some Spanish and Latin American genomes jumped between clusters corresponding to different subpopulations of hspSWEurope (Supplementary Figs. 3d and 4d). However, it was infrequent that genomes were reclassified across the main populations, which supported the relative stability of categories. A few genomes, especially from Indonesia, showed chimeric chromosome painting patterns, for example, a hspUral/hspEAsia combination and a hspUral/hpNEAfrica combination, which constitute rare and exciting intersects between distant populations.

The finding of a highly homogenous group of geographically dispersed genomes in the US motivated us to search for evidence of

distant clonal relationships amongst all *Hp*GP strains. The exceptional recombination rate of *H. pylori* means that strains with a common ancestor a few hundred years ago will have recombined most of their genomes, eliminating evidence of a shared clonal frame. Furthermore, an estimated 3.5 billion humans are infected with *H. pylori*[21] meaning that the current bacterial population size is enormous. As a result, it has been rare to find evidence of clonal relationships between strains collected from distant geographic locations. However, the availability of complete genomes makes it possible to detect deep clones that have recombined in a large fraction of their genome but still share some signal of clonal descent, and the probability of sampling clonally related strains increases quadratically with sample size, meaning that clones will become increasingly common as database sizes increase.

The frequency of the deep clone Hp_Clone_US-1 in the US population is likely somewhere between 3% (proportion in non-*Hp*GP US samples) and 18% (proportion in *Hp*GP), while it has not yet been found outside the US. The US population in the year 1830 was less than 13 million individuals and has increased to over 330 million through natural population growth and immigration. Assuming the lineage was introduced into the US by a single individual around 1830 and infected 10-fold or more people in each human generation, it would be present in around 3 million individuals today, or about 4% of sampled individuals. These calculations ignore factors such as mixed infection and are subject to many uncertainties but demonstrate that a high level of non-vertical transmission and a significant fitness advantage over other *H. pylori* is necessary to explain the current frequency of Hp_Clone_US-1 in US individuals.

The relative frequency of different transmission routes in the spread of *H. pylori* remains unclear, and while there is evidence of frequent vertical transmission in some populations, other evidence suggests the infection spreads more readily among children[22,23]. Recent work has emphasized the role of transmission within communities, especially in locations without modern sanitary infrastructure. Our results imply that Hp_Clone_US-1 has been expanding continuously, with several pairs of strains isolated from patients in different states having estimated common ancestors within the last 70 years, which suggests the possibility of occasional mass transmission events in the 20th-century USA. Identification of further clones worldwide should provide additional information to understand when and how some lineages of *H. pylori* can spread fast through human populations. Interestingly, all members of the clone lack the *cag* pathogenicity island, suggesting that also Cag negative strains can be highly competitive under modern conditions.

We note that several geographical regions and human populations remain understudied. Acquiring a better coverage of *H. pylori* whole genomes from South and Central Asia, and a broader representation from the Russian Federation is pivotal. These additional samples would not only offer deeper insights into the hspUral subclades but might also illuminate the possibility of uncovering novel subpopulations stemming from the main ancestral group, HpAsia2. Also, the African continent is still poorly studied in terms of *H. pylori* genomics, which severely limits our understanding of not only population structure but important aspects of bacterial virulence and pathophysiology.

This *Hp*GP manuscript was designed as a landmark paper, detailing *Helicobacter pylori* population structure in a global, high-quality dataset. Our intention is for the manuscript to serve as a launching point for individual researchers to deepen the exploration of the detailed data generated by our network. We hope the material (i.e., data and strains) generated by the *Hp*GP, including shared resources, codes, and interactive visualizations, together with our main results, will be widely used and will facilitate secondary analyses with the ultimate goal of reducing the burden of the pathologies associated with this bacterial carcinogen.

## Methods

### Sample acquisition

The *Hp*GP samples represent a convenient set. Contributors of samples were identified through advertisements at international scientific meetings, direct invitations to known colleagues and investigators with published sets of *H. pylori* strains, as well as referrals. A limited number of *H. pylori* genomes was publicly available from Spain, one of the main countries responsible for colonial activities in the Americas. Thus, in collaboration of members of the Spanish Association of Gastroenterology, we oversampled this country to better understand the admixed genomes from individuals from Latin America and the Caribbean.

We obtained gastric tissues (fresh frozen with and without culture media; $n = 351$) and cultures (pooled or single colonies; $n = 660$) of *H. pylori* from patients with non-atrophic gastritis ($n = 606$), advanced intestinal metaplasia ($n = 172$, with extension to gastric corpus or incomplete type restricted to antrum), and gastric cancer ($n = 233$). Samples were collected between 1995 and 2020. Biospecimens were shipped to the Division of Gastroenterology, Hepatology, and Nutrition at Vanderbilt University for processing. Before shipment, clinical information and sample descriptions were submitted to the coordinating center at the US National Cancer Institute to confirm eligibility. Biospecimens from the 72 collaborating centers were shipped frozen on dry ice. All individuals provided informed consent, and local Institutional Review Boards approved sample collection. The *Hp*GP was exempted from institutional review board evaluation by the National Institutes of Health Office of Human Subjects Research Protection. The summary statistics of 1011 included strains are presented in Table 1, and corresponding NCBI accession numbers and genome statistics are presented in Supplementary Data 1.

### Isolation and expansion of *H. pylori* strains and DNA extraction

Gastric tissues (biopsies or fragments from resections) were homogenized under sterile conditions in 100 μL of sterile phosphate-buffered saline (PBS, pH 7.4) using a homogenizer (Kimble–Kontes, Vineland, NJ, US). Then, 300 μL of sterile PBS was added to each sample, mixed, and plated onto two selective Trypticase soy agar (TSA) plates with 5% sheep blood containing vancomycin (20 mg/L), bacitracin (200 mg/L), nalidixic acid (10 mg/L) and amphotericin B (2 mg/L) (Sigma, St Louis, MO, US). In addition, a 1:10 dilution was plated on a no-antibiotic TSA plate (BBL; LABSCO, Nashville, TN, US). Agar plates were incubated under microaerobic conditions (Campy Pak Plus envelope, BBL) at 37 °C for 4–6 days until small gray translucent colonies appeared. Gram stains and assays for oxidase and urease were performed. Colony morphology was consistent with the characteristic shape of *H. pylori* colonies. A pool and one single colony of *H. pylori* were expanded and frozen into 1 mL of freezing media (Brucella broth plus 15% glycerol). The single colony was also expanded and used for DNA extraction using Qiagen, QIAamp DNA Mini kit (Qiagen, Catalog number 51306), following the protocol and using the EB buffer to elute the DNA. Original cultures (pooled or single colonies) were processed using the same protocol.

### PacBio whole-genome library preparation and sequencing

DNA samples were sequenced at the Cancer Genomics Research Laboratory at the US National Cancer Institute. The manufacturer's protocol was performed for constructing whole-genome libraries from microbial DNA using the SMRTbell Template Prep Kit. Briefly, 1000 ng of genomic DNA, as determined by Quant-iT™ PicoGreen dsDNA Reagent (Thermo Fisher Scientific, Waltham, MA, US), was sheared using the g-TUBE (Covaris, Inc., Woburn, MA, US) to an average fragment size of 10 kb. Following fragmentation and purification, DNA damage, and end repair, hairpin adapters were ligated to the fragment ends to generate SMRTbell libraries. For sequencing on the PacBio RSII instrument, standard hairpin adapters were used. For sequencing on

the PacBio Sequel and Sequel II instruments, barcoded hairpin adapters were used. For the Sequel and Sequel II, barcoded SMRTbell libraries were pooled (up to 8 for the Sequel and up to 48 for the Sequel II), and stringent purification was performed using AMPure PB beads to remove small fragments. Following purification, sequencing primer annealing and DNA polymerase binding of the pooled SMRTbell libraries was performed according to the manufacturer's protocols. SMRT sequencing of the libraries proceeded on the PacBio instrument using 1 SMRT Cell per isolate (RSII) or pool of libraries (Sequel, Sequel II). Genome coverage ranged from 111 to 5678× (median, 949×).

### Genome assembly

Since samples were sequenced from multiple generations of PacBio instruments (RSII, Sequel, Sequel II), raw data from the RSII in h5 format were converted to Sequel's subreads XML format so that the same analytical pipeline could be applied to data from all three instruments sequencer. Thus, RSII data were reanalyzed by the same analysis pipeline as Sequel and Sequel II after initial assembly by HGAP3. The original assemblies (from HGAP3) and the new assemblies from the newer SMRTlink assembly tools (HGAP4 or Microbial Assembly) were compared and were highly consistent. Newly reassembled RSII data that generated a single contiguous chromosomal contig were kept.

Whole-genome assembly using raw subreads was performed using SMRTLink's HGAP4/Microbial Assembly, as well as Hifiasm v0.13-r308[24] on the HiFi (circular consensus sequencing, CCS) reads. Prior to Hifiasm, raw subreads were converted to circular consensus reads, filtering for CCS reads with minimum predicted read quality higher than 0.99. Chromosomes and plasmids were assembled, which proved to be the most accurate and efficient to achieve complete assembly of all contigs in silico. Circularization with circlator v1.5.3[25] was performed on every contig in each strain. Bacterial chromosomal contig start points were all shifted to NusB gene with an additional 12 nt at the 3'-end. MUMmer v3.23[26] was used to perform a self-alignment to screen for assembly issues or artifactual contigs, and prokka v1.14.6[27] annotation of conserved *H. pylori* genes was run on both raw subread and HiFi read assemblies. The assemblies generated from raw subreads were generally used for methylation calling and downstream analysis unless they failed to be circularized, or prokka annotation suggested a pseudogene percentage higher than 5%, or they contained an unexpected number of tRNA/rRNAs. In those samples, the Hifiasm assembly was used. Candidate chromosomal contigs were also aligned to a published 26695 *H. pylori* strain (NC_000915.1) as a sanity check to verify the contig shared high homology with known *H. pylori*. The detailed analyses of *Hp*GP plasmid sequences and their geographic and chromosomal contexts will be reported in full elsewhere.

### Assembly quality control

To address the assembly quality, we applied the 3Cs protocol suggested by PacBio (https://www.pacb.com/blog/beyond-contiguity/). First, we assessed sequence contiguity and determined that the *Hp*GP de novo assemblies all have a contig N50 over 1 Mb. As expected, single chromosomal contigs range from ~1.5 to 1.7 Mb. Second, we measured the completeness of our assemblies using BUSCO (Benchmarking Universal Single-Copy Orthologs) scores[28] v5.1.3. BUSCO checks the presence or absence of highly conserved genes, and a score >95% is considered a good assembly. For the assemblies that did not achieve a BUSCO score as high as 95%, we either discarded that sequence, or a second attempt was carried out either in silico or in the laboratory. All 1011 *Hp*GP assemblies have BUSCO scores above 95%. To further measure correctness, we checked the ratio of pseudogenes, including frameshifted, incomplete, internal stop, ambiguous residues, and multiple problems against the total number of genes. Any assembly with a ratio of pseudogene of more than 5% of the total was discarded. Although the *Hp*GP set includes assemblies from three different PacBio

sequencing instruments (15 RSII, 832 Sequel, and 164 Sequel II), the measures of assembly quality (contiguity, BUSCO scores, genomic sequence length, number of total genes, and number of pseudogenes) were similar for the 1011 assemblies from these instruments.

Finally, a consolidated QC report was generated to summarize contig lengths, BUSCO score, and coverage depth (Supplementary Data 1). The minimal chromosomal contig average confidence QV score among most strains was as high as 90.

### National Center for Biotechnology Information (NCBI) annotation

The *Hp*GP chromosomal sequences were submitted to NCBI, including annotation with the NCBI Prokaryotic Genome Annotation Pipeline, PGAP (https://www.ncbi.nlm.nih.gov/genome/annotation_prok/)[29–31]. For sequences that could not be circularized ($n = 7$), 100 Ns were added to mark breakpoint locations in the genomic sequences. The individual accession numbers and genome statistics are presented in Supplementary Data 1.

### Representative genome dataset

To relate the *Hp*GP dataset to previous knowledge about *H. pylori* population structure, we used a reference dataset representing the 17 global *H. pylori* subpopulations ($n = 255$ genomes, see Supplementary Data 2) described prior to January 2022. To acquire a balanced dataset, we selected 15 genomes per subpopulation, and for the subpopulations with more reported genomes, we selected representatives based on (1) consistency of population assignments in previous publications, (2) assembly quality (contig number, genome since $1.7 \pm 0.2$ Mbp, and (3) as wide geographical representation within the subpopulation as possible to try to encompass the full breadth of each subpopulation. For the African continent, hpAfrica2, hspAfrica1SAfrica, hspAfrica1WAfrica, and hpNEAfrica were represented, and from Europe hspNEurope, hspSEurope, and hspSWEurope. From Asia, we complemented hpAsia2 and hspEAsia with newly published genomes from hpNorthAsia and the proposed subpopulations hspSiberia, and hspUral[14]. For the Americas hspSWEuropeLatinAmerica, hspAfrica1MiscAmericas, hspAfrica1NorthAmerica, hspIndigenousAmericaN, and hspIndigenousAmericaS were represented[8], and lastly, for Oceania we included hpSahul. The genomes were annotated using prokka v1.14.6[27] as previously described[8,11].

### Core genome analysis

All population structure analyses (fineSTRUCTURE, Chromosome Painting, Network analysis, and DAPC, were based on the same core gene alignment. This was generated using the prokka-annotated 1011 *Hp*GP genomes plus a resequenced ATCC reference strain 26695 (*Hp*GP-26695) and the 255 representative genomes, a total of 1267 genomes. The analysis was performed using the panaroo pipeline v1.2.10[32] using 90% protein sequence identity and 75% gene length coverage cut-off.

### Population structure analysis

The genome-wide haplotype data was calculated as described previously[33]: we conducted SNP calling for each alignment, and imputation for polymorphic sites with missing frequency <1% using BEAGLE v.3.3.2[34]. This genome-wide haplotype contained 387,927 SNPs in 1227 genes and was used to define isolate populations and subpopulations based on the similarity of the haplotype copying profiles obtained by fineSTRUCTURE v4. Then, fineSTRUCTURE[35] analysis was performed with 200,000 iterations of both the burn-in and Markov chain Monte Carlo (MCMC) method to cluster individuals based on the coancestry matrix as described[36]. The results were visualized as a heat map with each cell indicating the proportion of DNA "chunks" a recipient receives from each donor. Furthermore, the posterior distribution of the clusters was visualized using

fineSTRUCTURE's tree-building algorithm to define the populations and subpopulations produced. With the previously obtained coancestry matrix, multiple principal component analysis (PCA) was calculated to analyze the population structure in detail. Principal components (PCs) 1 to 11 were calculated and visualized using R.

## DAPC analysis

We employed discriminant analysis of principal components (DAPC) to further investigate the genetic structure of our data. DAPC describes clusters in genetic data by creating synthetic variables (discriminant functions) that maximize variance among groups while minimizing variance within groups[9]. DAPC is a multivariate approach, not model based; hence, it makes no assumptions about Hardy-Weinberg or linkage equilibrium on genetic loci. Before running the DAPC, we assessed the number of clusters most supported for our *H. pylori* dataset by employing the *find.clusters* function in adegenet R package, comparing the results of 100 independent runs using a custom-made R script and selecting the optimal number of clusters according to Bayesian information criteria (BIC). In order to assess the uncertainty of the group assignments of each individual, we visualized posterior group membership probabilities based on the DAPC analysis using the function compoplot.

Using SNP-sites v2.5.1[37], we extracted 601,000 SNPs from the 1267 genome panaroo core gene alignment. We employed the function *optim.a.score* of the adegenet R package to identify the optimal number of principal components to consider for the analyses, as too many could lead to overfitting, while a low number of components could decrease discriminatory power between groups.

We first ran a DAPC employing the most supported number of clusters/groups as estimated by the *find.cluster* analysis: $K = 6$. Initially, we ran the DAPC considering all the sequences in our alignment so as to visualize the entire genetic variability of our data. Given the outlier position of the two groups, we subsequently ran another DAPC analysis excluding these outliers to better emphasize the differences among the other clusters. Finally, we computed posterior group membership probability for each individual. This parameter is based on the retained discriminant functions of the DAPC analysis and represents the probability of each sample to be assigned to a group, which can be interpreted in order to assess how clear-cut or admixed the clusters are. We also ran the DAPC procedure considering $K = 17$, the same number of clusters identified by the fineSTRUCTURE analysis.

## Network analysis of core genes

The core gene alignment obtained with panaroo was used to estimate distances with PAUP[38] v4.0a166, using maximum likelihood criteria. Each distance was normalized between 0 and 1 as previously described[8]. With this normalization, 0 means the highest genetic similarity, and 1 signifies the highest dissimilarity between two strains. Next, a complete network is created, where all pairs of strains have a measure of genetic distance based on this previous normalization. Strains are represented as vertices, and their distances are represented as edges. In the beginning, this network is fully connected and has no perceptible structure. A process of edge and node pruning is carried out to reveal the underlying structure of the genetic similarity between strains. This process consists of ranking the values of the edges and removing them subsequently, starting with the most dissimilar (equal or close to one). This process is continued until the network is subdivided into a determined number of Connected Components (CC). We consider a CC as a set of more than two nodes connected between them but isolated from other groups of nodes. If a single node is stripped of all its edges (singleton), we discard this node from the set of nodes of the resulting network. Figure 2 was created following this pruning process with a CC threshold of 2. This means that edges and singletons were removed until the full network was separated into two groups of nodes, with the separated group being the hpAfrica2 group.

## Chromosome painting

**Full dataset analysis.** To identify the patterns of shared genomic content of *H. pylori* isolates, we conducted chromosome painting using ChromoPainterV2[35], designating all genomes as recipients (1011 *Hp*GP genomes), and randomly selected ~20 isolates per population as donors (335 genomes; Supplementary Data 3). Each strain was painted using all the other donor samples and the result is visualized in a bar plot built with R.

***Hp*GP only ancestral chromosome painting.** Since we have no genomes from the true historically ancestral populations, genomic ancestry is commonly inferred from contemporary representatives of these populations. A second chromosome painting was thus performed using hpAfrica2, hspAfrica1WAfrica, hpNEAfrica, hpAsia2, hpNorthAsia, hspUral, and hspEAsia as donors to infer ancestral contributions to the populations in the *Hp*GP dataset[3,10]. We also only selected donors among the reference collection for which *H. pylori* population assignment and geographical origin were concordant (Supplementary Data 3).

## Core gene multilocus sequence typing (cgMLST)

To investigate the existence of clonal relationships in *H. pylori*, we estimated the total number of identical loci shared among strains from the *Hp*GP dataset by performing a cgMLST as implemented by chewBBACA[39] software v2.8.5. chewBBACA uses a gene-by-gene method to compare coding sequences and assign alleles based on a BLAST Score Ratio (BSR)[40]. We first used Prodigal[41] v2.6.3, including the option -t to create a training file from the assembled version of the 26695 *H. pylori* reference strain resequenced as part of the *Hp*GP dataset. Then, the "CreateSchema" module of chewBBACA was applied to the 1011 *Hp*GP genomes and the Prodigal training file to estimate a whole-genome MLST (wgMLST) scheme. The 3943 wgMLST genes were then compared with the "AlleleCall" module, using the default BSR threshold of 0.6. A total of 867 genes identified as paralogs were removed from the wgMLST using the "RemoveGenes" module, reducing the scheme to 3076 loci. We then used the "ExtractCgMLST" module to create a cgMLST with all loci present in more than 95 percent of strains (--t 0.95), obtaining a total of 981,110 alleles for 1040 loci, an average of 943 different alleles per locus.

We last used the cgMLST allelic profile to calculate pairwise distances with GrapeTree[42] v1.5.0, running it in "--wgMLST" mode with the "distance" method (-method distance) while ignoring missing data (--missing 0). We analyzed the distribution of cgMLST distances between pairs of strains in categories such as "US clone", "US clone boundary", "US non-clone", "Chile", "Chilean hspSWEuropeChile", "non-Chilean hspSWEuropeChile", "within the same country", and "between different countries", as depicted in Fig. 4a.

## Analysis of public US genomes

We downloaded all whole-genome sequences publicly available in the EnteroBase *H. pylori* database (https://enterobase.warwick.ac.uk/species/index/helicobacter) with the US as the country of isolation as of September 18, 2022 ($n = 226$). Sixty-seven sequences were either isolated from non-human hosts, results of experimental infections, repeated samplings from the same individual or overlapping the *Hp*GP set, thus were excluded. The remaining 151 genomes (Supplementary Data 4) were combined with the *Hp*GP US genomes and the 255 references in a kmer-based genomic distance analysis using mash v2.3[43]. The five genomes clustering with the US deep clone were added to the dataset used for in-depth analysis.

## Dating of the US deep clone

A core gene alignment of the highly clonal US genomes, including the five public ones, was generated with panaroo using the settings described above. Three genomes, *Hp*GP-USA-401, *Hp*GP-USA-404, and

*Hp*GP-USA-414 had diverged from the clone both by phylogeny and chromosome painting profile and were excluded from further analysis. A phylogenetic tree was computed using PhyML v3.1[44] and input to ClonalFrameML v1.11-3-g4f13f23[45], executed using default parameters. Node ages were determined using the R BactDating package[46], using 10,000 Markov chain Monte Carlo iterations and a mutation rate of $1.38 \times 10^{-5}$ per site per year, as has previously been estimated[16].

## Data visualization

The map figures of the dataset's geographical distribution, including the gray background map, were plotted using the ggplot2[47] and ggmaps[48] package in R. The painting profiles were summarized as described above, and plotting and statistical analysis was performed in R using the ggplot2 and plotly[49] packages.

## Strain availability

The *Hp*GP set of *H. pylori* strains is available from the US National Cancer Institute for scientific purposes upon a reasonable request. However, restrictions apply to its availability as some samples require authorization from contributing centers to be distributed to third parties.

## Reporting summary

Further information on research design is available in the Nature Portfolio Reporting Summary linked to this article.

## Data availability

The whole-genome sequences generated within the *Hp*GP have been deposited in the NCBI GenBank database under BioProject accession code PRJNA529500 [https://www.ncbi.nlm.nih.gov/bioproject/PRJNA529500] (Supplementary Data 1). NCBI or equivalent public accessions for the reference set are listed in Supplementary Data 2. The whole *Hp*GP genome dataset and the 255 reference genomes are also deposited to Zenodo, DOI: 10.5281/zenodo.10048320. Source Data for the individual figures are available with this paper.

## Code availability

The computational scripts to process the data and plot figures are available at https://github.com/HpGP/Code-and-Data v1.0. This code is also archived on Zenodo under https://doi.org/10.5281/zenodo.8381170.

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

## Acknowledgements

Our special thanks are extended to all the individuals who were the hosts of the strains of the *Hp*GP collection, who represent the human populations that bear the burden of *H. pylori*-associated disease, and whose biological samples serve to advance research aimed at reducing this disease burden. We immensely thank Lisa D. Finkelstein, Ramona Bhattacharya, Jillian M. Varonin, Mary Jane Williams, Karen Williams Kinney, and Melissa A. Raymond from the US National Institutes of Health (National Cancer Institute's Technology Transfer Center, National Cancer Institute's Division of Cancer Epidemiology, and Genetics, and Division of Logistic Services) for their administrative and logistic support in establishing the collaboration agreements and importing the multiple sets of biospecimens. We also thank the US Centers for Disease Control and Prevention's Import Permit Program. We dedicate this work to our deceased colleagues Pablo Luna, Radislav Nakov, Bongani Kaimila, and Khean Lee Goh who passed away in recent years.

The *Hp*GP was mainly supported by the Intramural Research Program from the US National Cancer Institute (NCI), National Institutes of Health (NIH). This work was supported in part by the intramural research programs of the US National Library of Medicine, the US National Institute on Minority Health and Health Disparities, and the US National Institute of Allergy and Infectious Diseases. The members of the bioinformatics group received support from the Swedish Society for Medical Research (K.T.), Assar Gabrielsson Foundation (K.T.), and Magnus Bergvall Foundation (K.T.). The collaborating centers for sample collection received grant support from the US NIH (P01CA116087, R01CA077955, R01DK058587 and P30DK058404 to R.M.P.; P01CA028842 and R01CA190612 to K.T.W.; P01CA028842, R01CA190612, K07CA125588, R03CA167773 and P30CA068485 to D.R.M.; K08CA252635 to R.J.H., K22CA226395 to M.G.-P.; and U54GM133807 to M.C.-C.), the German Federal Ministry of Education and Research (BMBF-0315905D, ERA-NET PathoGenoMics to P.M.), the French Association pour la Recherche Contre le Cancer (8412 to F.M.), the French Institut National du Cancer (07/3D1616/IABC-23-12/NC-NG and 2014-152 to F.M.), the Canceropole Grand Sud-Ouest (2010-08-canceropole GSO-Universite Bordeaux 2 to F.M.), the Japanese National Institutes of Health (DK62813 to Y.Y.), the Japanese Ministry of Education, Culture, Sports, Science, and Technology (18KK0266, 19H03473, 21H00346 and 22H02871 to Y.Y.), the National Fund for Innovation and Development of Science and Technology from the Ministry of Higher Education Science and Technology of the Dominican Republic (2012-2013-2A1-65 and 2015-3A1-182 to M.C.), the National Cancer Center of South Korea (2210630, I.J.C.), ArcticNet (RES0010178 to K.J.G.), the Network of Centres of Excellence of Canada, the Canadian Institutes for Health Research (MOP115031 to K.J.G.), Alberta Innovates Health Solutions (201201159 to K.J.G.), the University of Malaya-Ministry of Higher Education (UM.C/625/1/HIR/MOHE/CHAN-02 to J.V.), the Ministry of Science and Technology of Vietnam, the Kyrgyz State Medical Academy, the Italian Ministry of Health for Institutional Research, the Chilean National Fund for Health Research and Development (FONIS A19/0188, FONDECYT 1230504 and ANID-FONDAP 152220002 to A.R.; CONICYT-FONDAP 15130011 and FONDECYT 1231773 to A.H.C.), the Chilean Cancer Prevention and Control Center, the Horizon 2020 Programme of European Union (825832, "CeLac and European consortium for a personalized medicine approach to Gastric Cancer," LEGACy, to T.F.-K. and A.R.), the Fundação de Amparo à Pesquisa do Estado de São Paulo (FAPESP; 2014/26847-0, 2018/14267-2, 2018/02972-3 to E.D.-N.), the Departamento de Ciência e Tecnologia (DECIT), Ministry of Health, Brazil (PRONON, SIPAR 2500.035-167/2015-23 to E.D.-N.), the Conselho Nacional de Desenvolvimento Científico e Tecnológico (CNPq, 314344/2020-9 to E.T.-S.), the Universidad de Costa Rica (742-B9-310 and 742-90912-19 to V.R.-M.), LABGIPAT (S.D.-B.), the Hospital Clínica Bíblica (C.C.-N.), the Greek Ministry of Culture and Education (InfeNeutra Project, NSRF 2007-2013, MIS450598, D.N.S.), the National Strategic Reference Framework Operational Program "Competitiveness, Entrepreneurship and Innovation" (NSRF 2014-2020, MIS5002486, D.N.S.), the Hellenic *Helicobacter pylori* Study Group (2012-2016, B.M.-G.), the Hellenic Society of Gastroenterology (National Multicenter Laboratory Surveillance Studies, 2018-2019, B.M.-G.), the Ministry of Science and Technology, Executive Yuan, Taiwan (109-2314-B-002-096; MOST 111-2314-B-002-012; MOST 109-2314-B-002-090-MY3 to J.-M.L. and M.-S.W.), the National Research Foundation of Singapore, the Singapore Ministry of Health's National Medical Research Council (Open Fund-Large Collaborative Grant, MOH-OFLCG18May-0003), the University of Puerto Rico Comprehensive Cancer Center, the Fondo Nacional de Desarrollo Científico y Tecnológico (196-2015-FONDECYT to C.C.), Universidad Científica del Sur, and Instituto Nacional de Enfermedades Neoplasicas (INEN, Peru).

The computations and data storage required for the analyses presented were enabled by resources in projects snic-2021/22-229 and snic-2021/23-234 provided by the National Academic Infrastructure for Supercomputing in Sweden (NAISS) and the Swedish National

Infrastructure for Computing (SNIC) at the UPPMAX HPC, partially funded by the Swedish Research Council through grant agreements 2022-06725, and 2018-05973.

## Author contributions

Manuscript conception and design: K.T. and Z.Y.M.R. Analysis supervision: K.T., S.G., and D.F. Data analysis: K.T., Z.Y.M.R., D.W., S.S.M., R.B.A., S.G., and R.C.T. Interpretation of results: K.T., Z.Y.M.R., S.S.M., R.B.A., S.G., R.C.T., D.F., M.C.C., and C.S.R. Manuscript writing: K.T., Z.Y.M.R., D.W., R.C.T., D.F., and M.C.C. Data coordinator: D.W. Editing of the manuscript: S.S.M., R.B.A., S.G., HpGP Research Network and C.S.R. Sample acquisition: HpGP Research Network. Conception and design of the HpGP initiative: M.C.C. and C.S.R. HpGP study coordinators: M.C.C. and C.S.R.

## Funding

## Competing interests

J.P.G. has served as a speaker, consultant, and advisory member for or has received research funding from Mayoly, Allergan, Diasorin, Gebro Pharma, and Richen. E.B.-M. has served as a speaker and consultant for Janssen, Chiesi, Kern and Takeda. R.M.F., J.C.M., and C.F. own patent WO/2018/169423 on microbiome markers for gastric cancer, and R.J.R. works for New England Biolabs, a company that sells research reagents, including restriction enzymes and DNA methyltransferases, to the scientific community. The remaining authors declare no competing interests.

## Additional information

[1]Department of Chemistry and Molecular Biology, University of Gothenburg, Gothenburg, Sweden. [2]Facultad de Ciencias Químicas, Universidad Autónoma de Chihuahua, Chihuahua, Chihuahua, México. [3]Cancer Genomics Research Laboratory, Frederick National Laboratory for Cancer Research, Frederick, MD, USA. [4]Division of Cancer Epidemiology and Genetics, National Cancer Institute, Rockville, MD, USA. [5]Instituto Nacional de Medicina Genómica, Ciudad de México, México. [6]Consejo Nacional de Ciencia y Tecnologia, Cátedras CONACYT, Ciudad de México, México. [7]Centro de Ciencias de la Complejidad, Universidad Nacional Autónoma de México, Ciudad de México, México. [8]Department of Life Sciences and Biotechnology, University of Ferrara, Ferrara, Italy. [9]Centre for Microbes Development and Health, Institute Pasteur Shanghai, Shanghai, China. [204]These authors contributed equally: Kaisa Thorell, Zilia Y. Muñoz-Ramírez. [205]These authors jointly supervised this work: M. Constanza Camargo, Charles S. Rabkin. ✉e-mail: kaisa.thorell@gu.se

## HpGP Research Network

Kaisa Thorell [1,204] ✉, Zilia Y. Muñoz-Ramírez [2,204], Difei Wang[3,4], Santiago Sandoval-Motta[5,6,7], Rajiv Boscolo Agostini[8], Silvia Ghirotto[8], Roberto C. Torres [9], Judith Romero-Gallo[10], Uma Krishna[10], Richard M. Peek Jr[10], M. Blanca Piazuelo[10], Naïma Raaf[11], Federico Bentolila[12], Hafeza Aftab[13], Junko Akada[14], Takashi Matsumoto[14], Freddy Haesebrouck[15], Rony P. Colanzi[16], Thais F. Bartelli[17], Diana Noronha Nunes[17], Adriane Pelosof[17], Claudia Zitron Sztokfisz[17], Emmanuel Dias-Neto[17], Paulo Pimentel Assumpção[18], Ivan Tishkov[19], Laure Brigitte Kouitcheu Mabeku[20], Karen J. Goodman[21], Janis Geary[21], Taylor J. Cromarty[21], Nancy L. Price[21], Douglas Quilty[22], Alejandro H. Corvalan[23], Carolina A. Serrano[24], Robinson Gonzalez[25], Arnoldo Riquelme[25], Apolinaria García-Cancino[26], Cristian Parra-Sepúlveda[26], Giuliano Bernal[27], Francisco Castillo[28], Alisa M. Goldstein[4], Nan Hu[4], Philip R. Taylor[4], Maria Mercedes Bravo[29], Alvaro Pazos[30], Luis E. Bravo[31], Keith T. Wilson[10], James G. Fox[32], Vanessa Ramírez-Mayorga[33], Silvia Molina-Castro[33], Sundry Durán-Bermúdez[34], Christian Campos-Núñez[35], Manuel Chaves-Cervantes[35], Evariste Tshibangu-Kabamba[36,37], Ghislain Disashi Tumba[37], Antoine Tshimpi-Wola[38], Patrick de Jesus Ngoma-Kisoko[38], Dieudonné Mumba Ngoyi[38,39], Modesto Cruz[40], Celso Hosking[40], José Jiménez Abreu[41], Christine Varon[42], Lucie Benejat[42], Ousman Secka[43], Alexander Link[44], Peter Malfertheiner[44], Michael Buenor Adinortey[45], Ansumana Sandy Bockarie[46], Cynthia Ayefoumi Adinortey[47], Eric Gyamerah Ofori[48], Dionyssios N. Sgouras[49], Beatriz Martinez-Gonzalez[49], Spyridon Michopoulos[50], Sotirios Georgopoulos[51], Elisa Hernandez[52], Braulio Volga Tacatic[53], Mynor Aguilar[54], Ricardo L. Dominguez[55], Douglas R. Morgan[56], Hjördís Harðardóttir[57], Anna Ingibjörg Gunnarsdóttir[57], Hallgrímur Guðjónsson[57], Jón Gunnlaugur Jónasson[57,58], Einar S. Björnsson[57,58], Mamatha Ballal[59], Vignesh Shetty[59,60],

Muhammad Miftahussurur[61], Titong Sugihartono[61], Ricky Indra Alfaray[61], Langgeng Agung Waskito[61], Kartika Afrida Fauzia[61], Ari Fahrial Syam[62], Hasan Maulahela[62], Reza Malekzadeh[63], Masoud Sotoudeh[63], Avi Peretz[64,65], Maya Azrad[64,65], Avi On[65,66], Valli De Re[67], Stefania Zanussi[67], Renato Cannizzaro[68], Vincenzo Canzonieri[69], Takaya Shimura[70], Kengo Tokunaga[71], Takako Osaki[71], Shigeru Kamiya[71], Khaled Jadallah[72], Ismail Matalka[72], Nurbek Igissinov[73], Mariia Satarovna Moldobaeva[74], Attokurova Rakhat[74], Il Ju Choi[75], Jae Gyu Kim[76], Nayoung Kim[77], Minkyo Song[4], Mārcis Leja[78,79,80], Reinis Vangravs[78], Ģirts Šķenders[78,80], Dace Rudzīte[80], Aiga Rūdule[78], Aigars Vanags[79], Ilze Kikuste[78], Juozas Kupcinskas[81], Jurgita Skieceviciene[81], Laimas Jonaitis[81], Gediminas Kiudelis[81], Paulius Jonaitis[81], Vytautas Kiudelis[81], Greta Varkalaite[81], Jamuna Vadivelu[82,83], Mun Fai Loke[82], Kumutha Malar Vellasamy[82], Roberto Herrera-Goepfert[84], Juan Octavio Alonso-Larraga[85], Than Than Yee[86], Kyaw Htet[86], Takeshi Matsuhisa[87], Pradeep Krishna Shrestha[88], Shamshul Ansari[89], Olumide Abiodun[90], Christopher Jemilohun[91], Kolawole Oluseyi Akande[92], Oluwatosin Olu-Abiodun[93], Francis Ajang Magaji[94], Ayodele Omotoso[95], Chukwuemeka Chukwunwendu Osuagwu[96], Uchenna Okonkwo[95], Opeyemi O. Owoseni[97], Carlos Castaneda[98], Miluska Castillo[99], Billie Velapatino[100], Robert H. Gilman[101], Paweł Krzyżek[102], Grażyna Gościniak[102], Dorota Pawełka[103], Izabela Korona-Glowniak[104], Halina Cichoz-Lach[105], Monica Oleastro[106], Ceu Figueiredo[107,108,109], Jose C. Machado[107,108,109], Rui M. Ferreira[107,108], Dmitry S. Bordin[110,111,112], Maria A. Livzan[113], Vladislav V. Tsukanov[114], Patrick Tan[115,116,117], Khay Guan Yeoh[118,119], Feng Zhu[118], Reid Ally[120,121], Rainer Haas[122], Milagrosa Montes[123], María Fernández-Reyes[123], Esther Tamayo[123], Jacobo Lizasoain[123], Luis Bujanda[124], Sergio Lario[125,126], María José Ramírez-Lázaro[125,126], Xavier Calvet[125,126], Eduard Brunet-Mas[125,126], María José Domper-Arnal[127,128], Sandra García-Mateo[127,128], Daniel Abad-Baroja[129], Pedro Delgado-Guillena[130], Leticia Moreira[131], Josep Botargues[132], Isabel Pérez-Martínez[133,134], Eva Barreiro-Alonso[133,135,136], Virginia Flores[137], Javier P. Gisbert[138,139], Edurne Amorena Muro[140], Pedro Linares[141], Vicente Martin[142], Laura Alcoba[141], Tania Fleitas-Kanonnikoff[143], Hisham N. Altayeb[144,145], Lars Engstrand[146], Helena Enroth[147], Peter M. Keller[148,149], Karoline Wagner[150], Daniel Pohl[151], Yi-Chia Lee[152], Jyh-Ming Liou[152], Ming-Shiang Wu[152], Bekir Kocazeybek[153], Suat Sarıbas[153], İhsan Tasçı[154], Süleyman Demiryas[154], Nuray Kepil[155], Luis Quiel[156], Miguel Villagra[157], Morgan Norton[158], Deborah Johnson[158], Robert J. Huang[159], Joo Ha Hwang[159], Wendy Szymczak[160], Saranathan Rajagopalan[160], Emmanuel Asare[160], William R. Jacobs Jr.[160], Haejin In[160,161], Roni Bollag[162], Aileen Lopez[162], Edward J. Kruse[163], Joseph White[163], David Y. Graham[164], Charlotte Lane[165], Yang Gao[165], Patricia I. Fields[165], Benjamin D. Gold[166], Marcia Cruz-Correa[167,168], María González-Pons[167], Luz M. Rodriguez[169], Vo Phuoc Tuan[170], Ho Dang Quy Dung[170], Tran Thanh Binh[170], Tran Thi Huyen Trang[171], Vu Van Khien[171], Xiongfong Chen[172], Castle Raley[173], Bailey Kessing[173], Yongmei Zhao[172], Bao Tran[173], Andrés J. Gutiérrez-Escobar[4], Yunhu Wan[3], Belynda Hicks[3], Bin Zhu[3], Kai Yu[4], Bin Zhu[4], Meredith Yeager[3], Amy Hutchinson[3], Kedest Teshome[3], Kristie Jones[3], Wen Luo[3], Quentin Jehanne[42], Yukako Katsura[174], Patricio Gonzalez-Hormazabal[175], Xavier Didelot[176], Sam Sheppard[177], Eduardo Tarazona-Santos[178], Leonardo Mariño-Ramírez[179], John T. Loh[10], Steffen Backert[180], Michael Naumann[181], Christian C. Abnet[4], Annemieke Smet[182], Douglas E. Berg[183], Álvaro Chiner-Oms[184,185], Iñaki Comas[185,186], Francisco José Martínez-Martínez[186], Roxana Zamudio[178,187], Philippe Lehours[42], Francis Megraud[42], Koji Yahara[188], Martin J. Blaser[189], Tamas Vincze[190], Richard D. Morgan[190], Richard J. Roberts[190], Stephen J. Chanock[4], John P. Dekker[191], Javier Torres[192], Timothy L. Cover[10,193], Mehwish Noureen[194], Wolfgang Fischer[122], Filipa F. Vale[195,196], Joshua L. Cherry[197,198], Naoki Osada[199], Masaki Fukuyo[200], Masanori Arita[201], Yoshio Yamaoka[14,164], Ichizo Kobayashi[202], Ikuo Uchiyama[203], Daniel Falush ©[9], M. Constanza Camargo ©[4,205] & Charles S. Rabkin[4,205]

[10]Department of Medicine, Vanderbilt University Medical Center, Nashville, TN, USA. [11]Department of Natural and Life Sciences, Faculty of Sciences, University of Algiers 1 Benyoucef Benkhedda, Algiers, Algeria. [12]Departamento de Medicina Interna, Hospital Alemán, Buenos Aires, Argentina. [13]Department of Gastroenterology, Dhaka Medical College and Hospital, Dhaka, Bangladesh. [14]Department of Environmental and Preventive Medicine, Oita University Faculty of Medicine, Yufu, Japan. [15]Department of Pathobiology, Pharmacology and Zoological Medicine, Faculty of Veterinary Medicine, Ghent University, Ghent, Belgium. [16]Hospital Universitario Japones, Santa Cruz de la Sierra, Bolivia. [17]A.C.Camargo Cancer Center, São Paulo, São Paulo, Brazil. [18]Núcleo de Pesquisas em Oncologia, Universidade Federal do Pará, Belém, Pará, Brazil. [19]Medical University of Sofia, Sofia, Bulgaria. [20]Department of Biochemistry, University of Dschang, Dschang, Cameroon. [21]Faculty of Medicine and Dentistry, Department of Medicine, University of Alberta, Edmonton, AB, Canada. [22]Queen's University, Kingston, ON, Canada. [23]Department of Hematology and Oncology, Faculty of Medicine, Pontificia Universidad Católica de Chile, Santiago, Chile. [24]Department of Pediatric Gastroenterology and Nutrition, Faculty of Medicine, Pontificia Universidad Católica de Chile, Santiago, Chile. [25]Department of Gastroenterology, Faculty of Medicine, Pontificia Universidad Católica de Chile, Santiago, Chile. [26]Facultad de Ciencias Biológicas, Universidad de Concepción, Concepción, Chile. [27]Cáncer Lab, Departamento de Ciencias Biomédicas, Facultad de Medicina, Universidad Católica del Norte (Coquimbo), Chile, Coquimbo, Chile. [28]Hospital Hanga Roa, Easter Island, Chile. [29]Grupo de Investigación en Biología del Cáncer, Instituto Nacional de Cancerología, Bogotá DC, Colombia. [30]Departamento de Biología, Universidad de Nariño, Pasto, Nariño, Colombia. [31]Escuela de Medicina, Universidad del Valle, Cali, Valle, Colombia. [32]Division of Comparative Medicine, Department of Biological Engineering, Massachusetts Institute of Technology, Cambridge, MA, USA. [33]Instituto de Investigaciones en Salud, Universidad de Costa Rica, San Jose, Costa Rica. [34]Laboratorio de Patología General y Gastrointestinal (LABGIPAT), San Jose, Costa Rica. [35]Servicio de Gastroenterología y Endoscopía Digestiva, Hospital Clínica Bíblica, San Jose, Costa Rica. [36]Faculty of Medicine, Osaka Metropolitan University, Osaka, Japan. [37]Faculty of Medicine, University of Mbuji-Mayi, Mbuji Mayi, Kasai-Oriental, Democratic Republic of the

Congo. [38]Faculty of Medicine, University of Kinshasa, Kinshasa, Democratic Republic of the Congo. [39]Department of Parasitology, National Institute of Biomedical Research, Kinshasa, Democratic Republic of the Congo. [40]Instituto de Microbiología y Parasitología, Universidad Autónoma de Santo Domingo, Santo Domingo, Dominican Republic. [41]Dominican-Japanese Digestive Disease Center, Dr Luis E. Aybar Health and Hygiene City, Santo Domingo, Dominican Republic. [42]Bordeaux Institute of Oncology, BRIC U1312, INSERM, Bordeaux, and National Reference Center for Campylobacters & Helicobacters, CHU de Bordeaux, Bordeaux, France. [43]Medical Research Council Unit, The Gambia at the London School of Hygiene & Tropical Medicine, Banjul, The Gambia. [44]Department of Gastroenterology, Hepatology and Infectious Diseases, Otto-von-Guericke University Magdeburg, Magdeburg, Germany. [45]Department of Biochemistry, School of Biological Sciences, University of Cape Coast, Cape Coast, Central Region, Ghana. [46]Department of Internal Medicine and Therapeutics, School of Medical Sciences, University of Cape Coast, Cape Coast, Central Region, Ghana. [47]Department of Molecular Biology and Biotechnology, School of Biological Sciences, University of Cape Coast, Cape Coast, Central Region, Ghana. [48]Department of Biology Education, Faculty of Science Education, University of Education, Winneba, Ghana. [49]Laboratory of Medical Microbiology, Hellenic Pasteur Institute, Athens, Greece. [50]Department of Gastroenterology, Alexandra Hospital, Athens, Greece. [51]Department of Gastroenterology, Athens Medical, P. Faliron Hospital, Athens, Greece. [52]Facultad de Ciencias Médicas, University of San Carlos of Guatemala, Guatemala City, Guatemala. [53]Unidad de Gastroenterología, Hospital Roosevelt, Guatemala City, Guatemala. [54]Gastrocentro, S.A., Guatemala City, Guatemala. [55]Departamento de Medicina Interna, Hospital de Occidente, Santa Rosa de Copán, Honduras. [56]School of Medicine, University of Alabama at Birmingham (UAB), Birmingham, AL, USA. [57]Landspítali – The National University Hospital of Iceland, Reykjavík, Iceland. [58]University of Iceland, Reykjavík, Iceland. [59]Department of Microbiology, Kasturba Medical College, Manipal Academy of Higher Education, Manipal, Karnataka, India. [60]Department of Medicine, University of Cambridge, Cambridge, UK. [61]Universitas Airlangga, Surabaya, East Java, Indonesia. [62]University of Indonesia, Jakarta, Indonesia. [63]Digestive Disease Research Institute, Tehran University of Medical Sciences, Tehran, Iran. [64]Clinical Microbiology Laboratory and Research Institute, Tzafon Medical Center, affiliated with Azrieli Faculty of Medicine, Bar Ilan University, Poriya, Israel. [65]Azrieli Faculty of Medicine, Bar Ilan University, Safed, Israel. [66]Pediatric Gastroenterology and Nutrition Unit, Tzafon Medical Center, Poriya, Israel. [67]Unit of Immunopathology and Oncological Biomarkers, Centro di Riferimento Oncologico di Aviano, Aviano, Italy. [68]Unit of Oncological Gastroenterology, Centro di Riferimento Oncologico di Aviano, Aviano, Italy. [69]Unit of Pathology, Centro di Riferimento Oncologico di Aviano, Aviano, Italy. [70]Department of Gastroenterology and Metabolism, Nagoya City University Graduate School of Medical Sciences, Nagoya, Japan. [71]School of Medicine, Kyorin University, Mitaka, Tokyo, Japan. [72]Jordan University of Science and Technology, Ar-Ramtha, Jordan. [73]Department of Surgical Diseases Internship, Astana Medical University, Nur-Sultan, Kazakhstan. [74]Kyrgyz State Medical Academy, Bishkek, Kyrgyzstan. [75]Center for Gastric Cancer, National Cancer Center, Goyang, South Korea. [76]Department of Internal Medicine, Chung-Ang University Hospital, Seoul, South Korea. [77]College of Medicine, Seoul National University, Seoul, South Korea. [78]Institute of Clinical and Preventive Medicine, University of Latvia, Riga, Latvia. [79]Digestive Diseases Centre GASTRO, Riga, Latvia. [80]Riga East University Hospital, Riga, Latvia. [81]Department of Gastroenterology, Institute for Digestive Research, Medical Academy, Lithuanian University of Health Sciences, Kaunas, Lithuania. [82]Department of Medical Microbiology, Faculty of Medicine, Universiti Malaya, Kuala Lumpur, Malaysia. [83]Medical Education Research and Development Unit, Faculty of Medicine, Universiti Malaya, Kuala Lumpur, Malaysia. [84]Departamento de Patología, Instituto Nacional de Cancerología, Mexico City, Mexico. [85]Servicio de Endoscopía, Instituto Nacional de Cancerología, Mexico City, Mexico. [86]Defence Services General Hospital, Yangon, Yangon, Yangon Region, Myanmar. [87]Nippon Medical School, Tokyo, Japan. [88]Department of Gastroenterology, Maharajgunj Medical Campus, Tribhuvan University Teaching Hospital, Kathmandu, Nepal. [89]Division of Health Sciences, Abu Dhabi Women's Campus, Higher Colleges of Technology, Abu Dhabi, United Arab Emirates. [90]Department of Community Medicine, Babcock University, Ilishan, Ogun State, Nigeria. [91]Department of Medicine, Babcock University, Ilishan, Ogun State, Nigeria. [92]Department of Medicine, College of Medicine, University of Ibadan, Ibadan, Oyo, Nigeria. [93]Department of Nursing, Crescent University, Abeokuta, Ogun State, Nigeria. [94]University Teaching Hospital, University of Jos, Jos, Plateau, Nigeria. [95]University of Calabar Teaching Hospital, Calabar, Cross River, Nigeria. [96]University of Nigeria Teaching Hospital, Ituku-Ozalla, Enugu State, Nigeria. [97]Department of Internal Medicine, Federal Medical Center Abeokuta, Abeokuta, Ogun State, Nigeria. [98]Faculty of Health Sciences, Universidad Científica del Sur, Lima, Peru. [99]Departamento de investigación, Instituto Nacional de Enfermedades Neoplasicas, Lima, Peru. [100]Universidad Peruana Cayetano Heredia, Lima, Peru. [101]Bloomberg School of Public Health, Johns Hopkins University, Baltimore, MD, USA. [102]Department of Microbiology, Wroclaw Medical University, Wrocław, Poland. [103]Department of Surgery Teaching, Wroclaw Medical University, Wrocław, Poland. [104]Department of Pharmaceutical Microbiology, Medical University of Lublin, Lublin, Poland. [105]Department of Gastroenterology with Endoscopic Unit, Medical University of Lublin, Lublin, Poland. [106]Instituto Nacional de Saúde Dr. Ricardo Jorge, Lisboa, Portugal. [107]Instituto de Patologia e Imunologia Molecular da Universidade do Porto, Porto, Portugal. [108]Instituto de Investigação e Inovação em Saúde, Universidade do Porto, Porto, Portugal. [109]Faculdade de Medicina da Universidade do Porto, Porto, Portugal. [110]Department of Pancreatic, Biliary and Upper Digestive Tract Disorders, A. S. Loginov Moscow Clinical Scientific Center, Moscow, Russia. [111]Department of General Medical Practice and Family Medicine, Tver State Medical University, Moscow, Russia. [112]Department of Propaedeutic of Internal diseases and Gastroenterology A.I. Yevdokimov Moscow State University of Medicine and Dentistry, Moscow, Russia. [113]Department of Faculty Therapy and Gastroenterology, Omsk State Medical University, Omsk, Russia. [114]Scientific Research Institute of Medical Problems of the North, Federal Research Centre "Krasnoyarsk Science Centre" of the Siberian Branch of Russian Academy of Science, Krasnoyarsk, Russia. [115]Cancer Science Institute of Singapore, National University of Singapore, Singapore, Singapore. [116]Cancer and Stem Cell Biology Program, Duke NUS Medical School, Singapore, Singapore. [117]Genome Institute of Singapore, Agency for Science, Technology and Research, Singapore, Singapore. [118]Department of Medicine, Yong Loo Lin School of Medicine, National University of Singapore, Singapore, Singapore. [119]Department of Gastroenterology and Hepatology, National University Health System, Singapore, Singapore. [120]Chris Hani Baragwanath Academic Hospital, Johannesburg, South Africa. [121]University of the Witwatersrand, Johannesburg, South Africa. [122]Max von Pettenkofer Institute of Hygiene and Medical Microbiology, Faculty of Medicine, LMU Munich, Munich, Germany. [123]Hospital Universitario Donostia, San Sebastian, Spain. [124]Department of Gastroenterology, Bioodonostia Health Research Institute - Donostia University Hospital, Universidad del País Vasco (UPV/EHU), Centro de Investigación Biomédica en Red de Enfermedades Hepáticas y Digestivas (CIBERehd), San Sebastian, Spain. [125]Digestive Diseases Unit, Parc Taulí Hospital Universitari. Institut d'Investigació i Innovació Parc Taulí (I3PT-CERCA), Universitat Autònoma de Barcelona, Barcelona, Spain. [126]Centro de Investigación Biomédica en Red de Enfermedades Hepáticas y Digestivas, Instituto de Salud Carlos III, Madrid, Spain. [127]Lozano Blesa University Clinic Hospital, Zaragoza, Spain. [128]Aragon Health Research Institute, Zaragoza, Spain. [129]Miguel Servet University Hospital, Zaragoza, Spain. [130]Hospital General de Granollers, Barcelona, Spain. [131]Gastroenterology Department, Hospital Clínic of Barcelona, Centro de Investigación Biomédica en Red de Enfermedades Hepáticas y Digestivas, Facultat de Medicina i Ciències de la Salut, Universitat de Barcelona, Barcelona, Spain. [132]Hospital Universitari de Bellvitge, L'Hospitalet de Llobregat, Barcelona, Spain. [133]Department of Gastroenterology, Hospital Universitario Central de Asturias, Oviedo, Asturias, Spain. [134]Diet, Microbiota and Health Group, Instituto de Investigación Sanitaria del Principado de Asturias, Oviedo, Asturias, Spain. [135]Farmacology Group, Instituto de Investigación Sanitaria del Principado de Asturias, Oviedo, Asturias, Spain. [136]Instituto Universitario de Oncología del Principado de Asturias, Oviedo, Asturias, Spain. [137]Hospital General Universitario Gregorio Marañón, Madrid, Spain. [138]Gastroenterology Unit, Hospital Universitario de La Princesa, Instituto de Investigación Sanitaria Princesa, Madrid, Spain. [139]Universidad Autónoma de Madrid, Centro de Investigación Biomédica en Red de Enfermedades Hepáticas y Digestivas, Madrid, Spain. [140]Gastroenterology

Department, Hospital Universitario de Navarra, Pamplona, Navarra, Spain. [141]Hospital de Leon, Leon, Spain. [142]Institut of Biomedicine, University of León, Consortium for Biomedical Research in Epidemiology and Public Health, Leon, Spain. [143]Instituto de Investigación Sanitaria INCLIVA, Hospital Clínico Universitario de Valencia, Valencia, Spain. [144]Biochemistry Department, Faculty of Science, King Abdulaziz University, Jeddah, Saudi Arabia. [145]Faculty of Medical laboratory Science, Sudan University of Science and Technology, Khartoum, Sudan. [146]Centre for Translational Microbiome Research, Department of Microbiology, Tumor and Cell Biology, Karolinska Institutet, Solna, Sweden. [147]University of Skövde, Skövde, Sweden. [148]Institute for Infectious Diseases, University of Bern, Bern, Switzerland. [149]Clinical Bacteriology/Mycology Unit, University Hospital Basel, Basel, Switzerland. [150]Institute of Medical Microbiology, University of Zurich, Zürich, Switzerland. [151]Clinic for Gastroenterology and Hepatology, University Hospital Zurich, Zürich, Switzerland. [152]College of Medicine, National Taiwan University, Taipei City, Taiwan. [153]Medical Microbiology Department, Cerrahpasa Medical Faculty, Istanbul University-Cerrahpasa, İstanbul, Türkiye. [154]General Surgery Department, Cerrahpasa Medical Faculty, Istanbul University-Cerrahpasa, İstanbul, Türkiye. [155]Medical Pathology Department, Cerrahpasa Medical Faculty, Istanbul University-Cerrahpasa, İstanbul, Türkiye. [156]Lawrence General Hospital, Lawrence, MA, USA. [157]Carson Tahoe Regional Medical Center, Carson City, NV, USA. [158]White River Medical Center, Batesville, AR, USA. [159]Department of Medicine, Stanford University, Stanford, CA, USA. [160]Albert Einstein College of Medicine, Bronx, NY, USA. [161]Rutgers Cancer Institute, New Brunswick, NJ, USA. [162]Georgia Cancer Center's Biorepository, Augusta University, Augusta, Georgia. [163]Augusta University Medical Center, Augusta, Georgia. [164]Section of Gastroenterology and Hepatology, Department of Medicine, Baylor College of Medicine, Houston, TX, USA. [165]Enteric Diseases Laboratory Branch, Division of Foodborne, Waterborne, and Environmental Diseases, Centers for Disease Control and Prevention, Atlanta, GA, USA. [166]Gi Care for Kids, LLC, Children's Center for Digestive Healthcare, LLC, Atlanta, GA, USA. [167]University of Puerto Rico Comprehensive Cancer Center, San Juan, Puerto Rico. [168]University of Puerto Rico Medical Sciences Campus, San Juan, Puerto Rico. [169]Gastrointestinal and Other Cancers Research Group, Division of Cancer Prevention, National Cancer Institute, Rockville, MD, USA. [170]Department of Endoscopy, Cho Ray Hospital, Ho Chi Minh City, Vietnam. [171]Department of Hepatogastroenterology, 108 Military Central Hospital, Hanoi, Vietnam. [172]Sequencing Facility Bioinformatics Group, Bioinformatics and Computational Science Directorate, Frederick National Laboratory for Cancer Research, Frederick, MD, USA. [173]Cancer Research Technology Program, Frederick National Laboratory for Cancer Research, Frederick, MD, USA. [174]Center for the Evolutionary Origins of Human Behavior, Kyoto University, Inuyama, Japan. [175]Instituto de Ciencias Biomédicas (ICBM), Facultad de Medicina, Universidad de Chile, Santiago, Chile. [176]School of Life Sciences, University of Warwick, Coventry, UK. [177]Department of Biology & Biochemistry, University of Bath, Bath, UK. [178]Departamento de Genética, Ecologia e Evolução, Universidade Federal de Minas Gerais, Belo Horizonte, Brazil. [179]National Institute on Minority Health and Health Disparities, Bethesda, MD, USA. [180]Division of Microbiology, Department of Biology, Friedrich-Alexander-Universität Erlangen-Nürnberg, Erlangen, Germany. [181]Institute of Experimental Internal Medicine, Otto-von-Guericke-Universität Magdeburg, Magdeburg, Germany. [182]Laboratory of Experimental Medicine and Pediatrics, Faculty of Medicine and Health Sciences, University of Antwerp, Antwerp, Belgium. [183]Department of Molecular Microbiology, Washington University School of Medicine, St. Louis, MO, USA. [184]Genomics and Health Area, FISABIO – Public Health, Valencia, Spain. [185]CIBER in Epidemiology and Public Health, Madrid, Spain. [186]Tuberculosis Genomics Unit, Instituto de Biomedicina de Valencia, Consejo Superior de Investigaciones Científicas, Valencia, Spain. [187]Quadram Institute Bioscience, Norwich, UK. [188]National Institute of Infectious Diseases, Tokyo, Japan. [189]Center for Advanced Biotechnology and Medicine, Rutgers University, New Brunswick, NJ, USA. [190]New England Biolabs, Ipswich, MA, USA. [191]Bacterial Pathogenesis and Antimicrobial Resistance Unit, National Institute of Allergy and Infectious Diseases, Bethesda, MD, USA. [192]Unidad de Investigación en Enfermedades Infecciosas y Parasitarias, UMAE Pediatría, Instituto de Seguro Social, Mexico City, Mexico. [193]Veterans Affairs Tennessee Valley Healthcare System, Nashville, TN, USA. [194]Department of Genetics, SOKENDAI University, Mishima, Shizuoka, Japan. [195]Pathogen Genome Bioinformatics and Computational Biology, Research Institute for Medicines, Faculty of Pharmacy, Universidade de Lisboa, Lisboa, Portugal. [196]Instituto de Biosistemas e Ciências Integrativas, Faculdade de Ciências, Universidade de Lisboa, Lisboa, Portugal. [197]National Center for Biotechnology Information, National Library of Medicine, National Institutes of Health, Bethesda, MD, USA. [198]Division of International Epidemiology and Population Studies, Fogarty International Center, National Institutes of Health, Bethesda, MD, USA. [199]Faculty of Information Science and Technology, Hokkaido University, Sapporo, Japan. [200]Department of Molecular Oncology, Chiba University, Chiba, Japan. [201]Bioinformation and DDBJ Center, National Institute of Genetics, Mishima, Shizuoka, Japan. [202]Research Center for Micro-Nano Technology, Hosei University, Tokyo, Japan. [203]National Institute for Basic Biology, National Institutes of Natural Sciences, Aichi, Japan.

