## [Peer Review File · Nature Communications]

The Helicobacter pylori Genome Project: insights into H. pylori population structure from analysis of a worldwide collection of complete genomesEditorial Note: This manuscript has been previously reviewed at another journal that is not operating a transparent peer review scheme. This document only contains reviewer comments and rebuttal letters for versions considered at *Nature Communications*.

Reviewer #3 (Remarks to the Author):

The authors have addressed most of my concerns except for the following:

1. NB: this is a non-scientific criticism. L435 "took the vanguard" (to my mind) frames the imperialism of Spain rather positively - the authors may wish to revise their language here. e.g. 'responsible for building many of the early colonial societies'
2. While I appreciate the efforts to better outline the sample accumulation strategy, the authors state they oversampled Spain and then in the response to justifying the US clone's importance state 'They are confident they did not oversample any region' (presumably except Spain?)
3. Regarding Figure 4a - this hasn't been changed at all. The issue is that, I believe (though the Figure is hard to understand so I could be mistaken), there is double-counting of pairs. e.g. a pairwise cgMLST distance between two hspSWEuChile isolates is represented in both pale blue 'hspSWEuChile vs hspSWEuChile' and 'within the same country'. An alternative, clearer representation would be to have a series of stacked bar charts with individual x-axes which would allow meaningful comparisons to be made across the distributions.

Also, the labels need to be updated since the authors have changed the name of 'the US clone'.

And the updated figure legend says lower values 'represent greater similarity'. This is clearly a proportion of something - can it not just be stated what it is a proportion of for clarity? e.g. shared/unshared MLST alleles? or at least added to the methods?

Response to reviewers

Reviewer #3:

1. NB: this is a non-scientific criticism. L435 "took the vanguard" (to my mind) frames the imperialism of Spain rather positively - the authors may wish to revise their language here. e.g. 'responsible for building many of the early colonial societies'

We agree that this is a delicate question with different perceptions in different parts of the world and should be rephrased. We suggest "one of the main countries responsible for colonial activities in the Americas".

2. While I appreciate the efforts to better outline the sample accumulation strategy, the authors state they oversampled Spain and then in the response to justifying the US clone's importance state 'They are confident they did not oversample any region' (presumably except Spain?)

We agree with the reviewer that this statement was unclear. What we meant to highlight was that the discovery of the deep clone was not merely due to sampling of a specific region of the US. While for most countries the samples were collected at one or a few hospitals or cities, in the case of the US they were actually from several collaborating partners leading to a wide geographical distribution.

3. Regarding Figure 4a - this hasn't been changed at all. The issue is that, I believe (though the Figure is hard to understand so I could be mistaken), there is double-counting of pairs. e.g. a pairwise cgMLST distance between two hspSWEuChile isolates it represented in both pale blue 'hspSWEuChile vs hspSWEuChile' and 'within the same country'. An alternative, clearer representation would be to have a series of stacked bar charts with individual x-axes which would allow meaningful comparisons to be made across the distributions.

Also, the labels need to be updated since the authors have changed the name of 'the US clone'.

And the updated figure legend says lower values 'represent greater similarity'. This is clearly a proportion of something - can it not just be stated what it is a proportion of for clarity? e.g. shared/unshared MLST alleles? or at least added to the methods?

The concern of the reviewer, that some pairwise comparisons were represented twice was indeed correct and we have made modifications to the figure to eliminate double counts. As a result, pairs that meet the criteria of the categories of interest are no longer included in other categories such as "Other pairs within the same country" and "Other pairs between different countries."

We have also discussed the suggested option of creating series of stacked bar charts. However, with the double counting issue resolved, we believe that this representation

remains the most effective way to compare the fraction of pairs in each category across the distribution.

In addition we addressed an issue of truncated data in the previous version of the figure that we haven't noticed and have revised the labels and legend to clarify these points. We also included the new term for the clone both in the figure legend and in the text (L296).

Reviewer #3 (Remarks to the Author):

The authors have addressed my concerns.